# High-Dimensional Optimization in Adaptive Random Subspaces

**Jonathan Lacotte**
Department of Electrical Engineering
Stanford University
lacotte@stanford.edu

**Mert Pilanci**
Department of Electrical Engineering
Stanford University

**Marco Pavone**
Department of Aeronautics &Astronautics
Stanford University

## Abstract

We propose a new randomized optimization method for high-dimensional problems which can be seen as a generalization of coordinate descent to random subspaces. We show that an adaptive sampling strategy for the random subspace significantly outperforms the oblivious sampling method, which is the common choice in the recent literature. The adaptive subspace can be efficiently generated by a correlated random matrix ensemble whose statistics mimic the input data. We prove that the improvement in the relative error of the solution can be tightly characterized in terms of the spectrum of the data matrix, and provide probabilistic upper-bounds. We then illustrate the consequences of our theory with data matrices of different spectral decay. Extensive experimental results show that the proposed approach offers significant speed ups in machine learning problems including logistic regression, kernel classification with random convolution layers and shallow neural networks with rectified linear units. Our analysis is based on convex analysis and Fenchel duality, and establishes connections to sketching and randomized matrix decomposition.

## 1 Introduction

Random Fourier features, Nystrom method and sketching techniques have been successful in large scale machine learning problems. The common practice is to employ oblivious sampling or sketching matrices, which are typically randomized and fixed ahead of the time. However, it is not clear whether one can do better by adapting the sketching matrices to data. In this paper, we show that adaptive sketching matrices can significantly improve the approximation quality. We characterize the approximation error on the optimal solution in terms of the smoothness of the function, and spectral properties of the data matrix.

Many machine learning problems end up being high dimensional optimization problems, which typically follow from forming the kernel matrix of a large dataset or mapping the data trough a high dimensional feature map, such as random Fourier features [20] or convolutional neural networks [13]. Such high dimensional representations induce higher computational and memory complexities, and result in slower training of the models. Random projections are a classical way of performing dimensionality reduction, and are widely used in many algorithmic contexts [25]. Nevertheless, only recently these methods have captured great attention as an effective way of performing dimensionality reduction in convex optimization. In the context of solving a linear system $Ax = b$ and least-squares optimization, the authors of [14] propose a randomized iterative method with linear convergence rate,

which, at each iteration, performs a proximal update $x^{(k+1)} = \mathrm{argmin}_{x \in T} \|x - x^{(k)}\|_2^2$, where the next iterate $x^{(k+1)}$ is restricted to lie within an affine subspace $T = x^{(k)} + \mathrm{range}(A^\top S)$, and $S$ is a $n \times m$ dimension-reduction matrix with $m \leqslant \min\{n, d\}$. In the context of kernel ridge regression, the authors of [31] propose to approximate the $n$-dimensional kernel matrix by sketching its columns to a lower $m$-dimensional subspace, chosen uniformly at random. From the low dimensional kernel ridge solution $\alpha^* \in \mathbb{R}^m$, they show how to reconstruct an approximation $\widetilde{x} \in \mathbb{R}^n$ of the high dimensional solution $x^* \in \mathbb{R}^n$. Provided that the sketching dimension $m$ is large enough – as measured by the spectral properties of the kernel matrix $K$ –, the estimate $\widetilde{x}$ retains some statistical properties of $x^*$, e.g., minimaxity. Similarly, in the broader context of classification through convex loss functions, the authors of [32, 33] propose to project the $d$-dimensional features of a given data matrix $A$ to a lower $m$-dimensional subspace, chosen independently of the data. After computing the optimal low-dimensional classifier $\alpha^* \in \mathbb{R}^m$, their algorithm returns an estimate $\widetilde{x} \in \mathbb{R}^d$ of the optimal classifier $x^* \in \mathbb{R}^d$. Even though they provide formal guarantees on the estimation error $\|\widetilde{x} - x^*\|_2$, their results rely on several restrictive assumptions, that is, the data matrix $A$ must be low rank, or, the classifier $x^*$ must lie in the span of the top few left singular vectors of $A$. Further, random subspace optimization has also been explored for large-scale trust region problems [27], also using a subspace chosen uniformly at random. Our proposed approach draws connections with the Gaussian Kaczmarz method proposed in [14] and the kernel sketching method in [31]. Differently, we are interested in smooth, convex optimization problems with ridge regularization. In contrast to [32, 33], we do not make any assumption on the optimal solution $x^*$.

Our work relates to the considerable amount of literature on randomized approximations of high dimensional kernel matrices $K$. The typical approach consists of building a low-rank factorization of the matrix $K$, using a random subset of its columns [28, 23, 9, 17]. The so-called Nystrom method has proven to be effective empirically [30], and many research efforts have been devoted to improving and analyzing the performance of its many variants (e.g., uniform column sub-sampling, leverage-score based sampling), especially in the context of kernel ridge regression [1, 2]. In a related vein, sketching methods have been proposed to reduce the space complexity of storing high-dimensional data matrices [19, 31], by projecting their rows to a randomly chosen lower dimensional subspace. Our theoretical findings build on known results for low-rank factorization of positive semi-definite (p.s.d.) matrices [15, 5, 12, 29], and show intimate connections with kernel matrices sketching [31]. Lastly, our problem setting also draws connections with compressed sensing [7] where the goal is to recover a high dimensional structured signal from a small number of randomized, and usually oblivious, measurements.

## 1.1 Contributions

In this work, we propose a novel randomized subspace optimization method with strong solution approximation guarantees which outperform oblivious sampling methods. We derive probabilistic bounds on the error of approximation for general convex functions. We show that our method provides a significant improvement over the oblivious version, and theoretically quantify this observation as function of the spectral properties of the data matrix. We also introduce an iterative version of our method, which converges to the optimal solution by iterative refinement.

## 1.2 An overview of our results

Let $f : \mathbb{R}^n \to \mathbb{R}$ be a convex and $\mu$-strongly smooth function, i.e., $\nabla^2 f(w) \preceq \mu I_n$ for all $w \in \mathbb{R}^n$, and $A \in \mathbb{R}^{n \times d}$ a high-dimensional matrix. We are interested in solving the *primal* problem

$$x^* = \underset{x \in \mathbb{R}^d}{\mathrm{argmin}}\, f(Ax) + \frac{\lambda}{2} \|x\|_2^2 \,, \tag{1}$$

Given a random matrix $S \in \mathbb{R}^{d \times m}$ with $m \ll d$, we consider instead the *sketched primal* problem

$$\alpha^* \in \underset{\alpha \in \mathbb{R}^m}{\mathrm{argmin}}\, f(AS\alpha) + \frac{\lambda}{2} \alpha^\top S^\top S \alpha \,, \tag{2}$$

where we effectively restrict the optimization domain to a lower $m$-dimensional subspace. In this work, we explore the following questions: How can we estimate the original solution $x^*$ given the sketched solution $\alpha^*$? Is a uniformly random subspace the optimal choice, e.g., $S \sim$ Gaussian i.i.d.?

Or, can we come up with an adaptive sampling distribution that is related to the matrix $A$, which yields stronger guarantees?

By Fenchel duality analysis, we exhibit a natural candidate for an approximate solution to $x^*$, given by $\widetilde{x} = -\lambda^{-1}A^\top\nabla f(AS\alpha^*)$. Our main result (Section 2) establishes that, for an *adaptive* sketching matrix of the form $S = A^\top\widetilde{S}$ where $\widetilde{S}$ is typically Gaussian i.i.d., the relative error satisfies a high-probability guarantee of the form $\|\widetilde{x} - x^*\|_2 / \|x^*\|_2 \leqslant \varepsilon$, with $\varepsilon < 1$. Our error bound $\varepsilon$ depends on the smoothness parameter $\mu$, the regularization parameter $\lambda$, the shape of the domain of the Fenchel conjugate $f^*$ and the spectral decay of the matrix $A$. Further, we show that this error can be explicitly controlled in terms of the singular values of $A$, and we derive concrete bounds for several standard spectral profiles, which arise in data analysis and machine learning. In particular, we show that using the *adaptive* matrix $S = A^\top\widetilde{S}$ provides much stronger guarantees than *oblivious* sketching, where $S$ is independent of $A$. Then, we take advantage of the error contraction (i.e., $\varepsilon < 1$), and extend our adaptive sketching scheme to an iterative version (Section 3), which, after $T$ iterations, returns a higher precision estimate $\widetilde{x}^{(T)}$ that satisfies $\|\widetilde{x}^{(T)} - x^*\|_2 / \|x^*\|_2 \leqslant \varepsilon^T$. Throughout this work, we specialize our formal guarantees and empirical evaluations (Section 5) to Gaussian matrices $\widetilde{S}$, which is a standard choice and yields the tightest error bounds. However, our approach extends to a broader class of matrices $\widetilde{S}$, such as Rademacher matrices, sub-sampled randomized Fourier (SRFT) or Hadamard (SRHT) transforms, and column sub-sampling matrices. Thus, it provides a general framework for random subspace optimization with strong solution guarantees.

## 2 Convex optimization in adaptive random subspaces

We introduce the Fenchel conjugate of $f$, defined as $f^*(z) := \sup_{w \in \mathbb{R}^n} \{w^\top z - f(w)\}$, which is convex and its domain $\mathrm{dom}f^* := \{z \in \mathbb{R}^n \mid f^*(z) < +\infty\}$ is a closed, convex set. Our control of the relative error $\|\widetilde{x} - x^*\|_2 / \|x^*\|_2$ is closely tied to controlling a distance between the respective solutions of the dual problems of (1) and (2). The proof of the next two Propositions follow from standard convex analysis arguments [22], and are deferred to Appendix C.

**Proposition 1** (Fenchel Duality). *Under the previous assumptions on $f$, it holds that*

$$\min_x f(Ax) + \frac{\lambda}{2}\|x\|_2^2 = \max_z -f^*(z) - \frac{1}{2\lambda}\|A^\top z\|_2^2.$$

*There exist an unique primal solution $x^*$ and an unique dual solution $z^*$. Further, we have $Ax^* \in \partial f^*(z^*)$, $z^* = \nabla f(Ax^*)$ and $x^* = -\frac{1}{\lambda}A^\top z^*$.*

**Proposition 2** (Fenchel Duality on Sketched Program). *Strong duality holds for the sketched program*

$$\min_\alpha f(AS\alpha) + \frac{\lambda}{2}\|S\alpha\|_2^2 = \max_y -f^*(y) - \frac{1}{2\lambda}\|P_S A^\top y\|_2^2,$$

*where $P_S = S(S^\top S)^\dagger S^\top$ is the orthogonal projector onto the range of $S$. There exist a sketched primal solution $\alpha^*$ and an unique sketched dual solution $y^*$. Further, for any solution $\alpha^*$, it holds that $AS\alpha^* \in \partial f^*(y^*)$ and $y^* = \nabla f(AS\alpha^*)$.*

We define the following deterministic functional $Z_f$ which depends on $f^*$, the data matrix $A$ and the sketching matrix $S$, and plays an important role in controlling the approximation error,

$$Z_f \equiv Z_f(A, S) = \sup_{\Delta \in (\mathrm{dom}f^* - z^*)} \left(\frac{\Delta^\top AP_S^\perp A^\top \Delta}{\|\Delta\|_2^2}\right)^{\frac{1}{2}}, \tag{3}$$

where $P_S^\perp = I - P_S$ is the orthogonal projector onto $\mathrm{range}(S)^\perp$. The relationship $x^* = -\lambda^{-1}A^\top\nabla f(Ax^*)$ suggests the point $\widetilde{x} = -\lambda^{-1}A^\top\nabla f(AS\alpha^*)$ as a candidate for approximating $x^*$. The Fenchel dual programs of (1) and (2) only differ in their quadratic regularization term, $\|A^\top z\|_2^2$ and $\|P_S A^\top y\|_2^2$, which difference is tied to the quantity $\|P_S^\perp A^\top(z - y)\|_2$. As it holds that $\|\widetilde{x} - x^*\|_2 = \lambda^{-1}\|A^\top(z^* - y^*)\|_2$, we show that the error $\|\widetilde{x} - x^*\|_2$ can be controlled in terms of the spectral norm $\|P_S^\perp A^\top\|_2$, or more sharply, in terms of the quantity $Z_f$, which satisfies $Z_f \leqslant \|P_S^\perp A^\top\|_2$. We formalize this statement in our next result, which proof is deferred to Appendix B.1.

**Theorem 1** (Deterministic bound). *Let $\alpha^*$ be any minimizer of the sketched program* (2). *Then, under the condition $\lambda \geqslant 2\mu Z_f^2$, we have*

$$\|\widetilde{x} - x^*\|_2 \leqslant \sqrt{\frac{\mu}{2\lambda}} Z_f \|x^*\|_2 \,, \tag{4}$$

*which further implies*

$$\|\widetilde{x} - x^*\|_2 \leqslant \sqrt{\frac{\mu}{2\lambda}} \|P_S^\perp A^\top\|_2 \|x^*\|_2 \,. \tag{5}$$

For an adaptive sketching matrix $S = A^\top \widetilde{S}$, we rewrite $\|P_S^\perp A^\top\|_2^2 = \|K - K\widetilde{S}(\widetilde{S}^\top K \widetilde{S})^\dagger \widetilde{S}^\top K\|_2$, where $K = AA^\top$ is p.s.d. Combining our deterministic bound (5) with known results [15, 12, 5] for randomized low-rank matrix factorization in the form $K\widetilde{S}(\widetilde{S}^\top K \widetilde{S})^\dagger \widetilde{S}^\top K$ of p.s.d. matrices $K$, we can give guarantees with high probability (w.h.p.) on the relative error for various types of matrices $\widetilde{S}$. For conciseness, we specialize our next result to adaptive Gaussian sketching, i.e., $\widetilde{S}$ Gaussian i.i.d. Given a target rank $k \geqslant 2$, we introduce a measure of the spectral tail of $A$ as $R_k(A) = \left(\sigma_k^2 + \frac{1}{k}\sum_{j=k+1}^\rho \sigma_j^2\right)^{\frac{1}{2}}$, where $\rho$ is the rank of the matrix $A$ and $\sigma_1 \geqslant \sigma_2 \geqslant \ldots \geqslant \sigma_\rho$ its singular values. The proof of the next result follows from a combination of Theorem 1 and Corollary 10.9 in [15], and is deferred to Appendix B.2.

**Corollary 1** (High-probability bound). *Given $k \leqslant \min(n,d)/2$ and a sketching dimension $m = 2k$, let $S = A^\top \widetilde{S}$, with $\widetilde{S} \in \mathbb{R}^{n \times m}$ Gaussian i.i.d. Then, for some universal constant $c_0 \leqslant 36$, provided $\lambda \geqslant 2\mu c_0^2 R_k^2(A)$, it holds with probability at least $1 - 12e^{-k}$ that*

$$\|\widetilde{x} - x^*\|_2 \leqslant c_0 \sqrt{\frac{\mu}{2\lambda}} R_k(A) \|x^*\|_2 \,. \tag{6}$$

**Remark 1.** *The quantity $Z_f := \sup_{\Delta \in (dom f^* - z^*)} \left(\frac{\Delta^\top A P_S^\perp A^\top \Delta}{\|\Delta\|_2^2}\right)^{\frac{1}{2}}$ is the eigenvalue of the matrix $P_S^\perp A^\top$, restricted to the spherical cap $\mathcal{K} := (dom f^* - z^*) \cap \mathcal{S}^{n-1}$, where $\mathcal{S}^{n-1}$ is the unit sphere in dimension $n$. Thus, depending on the geometry of $\mathcal{K}$, the deterministic bound (4) might be much tighter than (5), and yield a probabilistic bound better than (6). The investigation of such a result is left for future work.*

### 2.1 Theoretical predictions as a function of spectral decay

We study the theoretical predictions given by (5) on the relative error, for different spectral decays of $A$ and sketching methods, in particular, adaptive Gaussian sketching versus oblivious Gaussian sketching and leverage score column sub-sampling [12]. We denote $\nu_k = \sigma_k^2$ the eigenvalues of $AA^\top$. For conciseness, we absorb $\mu$ into the eigenvalues by setting $\nu_k \equiv \mu\nu_k$ and $\mu \equiv 1$. This re-scaling leaves the right-hand side of the bound (5) unchanged, and does not affect the analysis below. Then, we assume that $\nu_1 = \mathcal{O}(1)$, $\lambda \in (\nu_\rho, \nu_1)$, and $\lambda \to 0$ as $n \to +\infty$. These assumptions are standard in empirical risk minimization and kernel regression methods [11], which we focus on in Sections 4 and 5. We consider three decaying schemes of practical interest. The matrix $A$ has either a finite-rank $\rho$, a $\kappa$-exponential decay where $\nu_j \sim e^{-\kappa j}$ and $\kappa > 0$, or, a $\beta$-polynomial decay where $\nu_k \sim j^{-2\beta}$ and $\beta > 1/2$. Among other examples, these decays are characteristic of various standard kernel functions, such as the polynomial, Gaussian and first-order Sobolev kernels [3]. Given a precision $\varepsilon > 0$ and a confidence level $\eta \in (0,1)$, we denote by $m_A$ (resp. $m_O$, $m_S$) a sufficient dimension for which adaptive (resp. oblivious, leverage score) sketching yields the following $(\varepsilon, \eta)$-guarantee on the relative error. That is, with probability at least $1 - \eta$, it holds that $\|\widetilde{x} - x^*\|_2 / \|x^*\|_2 \leqslant \varepsilon$.

We determine $m_A$ from our probabilistic regret bound (6). For $m_S$, using our deterministic regret bound (5), it then suffices to bound the spectral norm $\|P_{A^\top \widetilde{S}}^\perp A^\top\|_2$ in terms of the eigenvalues $\nu_k$, when $\widetilde{S}$ is a leverage score column sub-sampling matrix. To the best of our knowledge, the tightest bound has been given by [12] (see Lemma 5). For $m_O$, we leverage results from [32]. The authors provide an upper bound on the relative error $\|\widetilde{x} - x^*\|_2 / \|x^*\|_2$, when $S$ is Gaussian i.i.d. with variance $\frac{1}{d}$. It should be noted that their sketched solution $\alpha^*$ is slightly different from ours. They solve $\alpha^* = \operatorname{argmin} f(AS\alpha) + (2\lambda)^{-1}\|\alpha\|_2^2$, whereas we do include the matrix $S$ in the regularization

term. One might wonder which regularizer works best when $S$ is Gaussian i.i.d. Through extensive numerical simulations, we observed a strongly similar performance. Further, standard Gaussian concentration results yields that $\|S\alpha^*\|_2^2 \approx \|\alpha^*\|_2^2$.

Our theoretical findings are summarized in Table 1, and we give the mathematical details of our derivations in Appendix D. For the sake of clarity, we provide in Table 1 lower bounds on the predicted values $m_O$ and $m_S$, and, thus, lower bounds on the ratios $m_O/m_A$ and $m_S/m_A$. Overall, adaptive Gaussian sketching provides stronger guarantees on the relative error $\|\widetilde{x} - x^*\|_2/\|x^*\|_2$.

Table 1: Sketching dimensions for a $(\varepsilon, \eta)$-guarantee on the relative error $\|\widetilde{x} - x^*\|_2/\|x^*\|_2$.

| | $\rho$-rank matrix ($\rho \ll n \wedge d$) | $\kappa$-exponential decay ($\kappa > 0$) | $\beta$-polynomial decay ($\beta > 1/2$) |
|---|---|---|---|
| Adaptive Gaussian ($m_A$) | $\rho + 1 + \log\left(\frac{12}{\eta}\right)$ | $\kappa^{-1}\log\left(\frac{1}{\lambda\varepsilon}\right) + \log\left(\frac{12}{\eta}\right)$ | $\lambda^{-1/2\beta}\varepsilon^{-1/\beta} + \log\left(\frac{12}{\eta}\right)$ |
| Oblivious Gaussian ($m_O$) | $(\rho+1)\varepsilon^{-2}\log\left(\frac{2\rho}{\eta}\right)$ | $\kappa^{-1}\varepsilon^{-2}\log\left(\frac{1}{\lambda}\right)\log\left(\frac{2d}{\eta}\right)$ | $\lambda^{-\frac{1}{2\beta}}\varepsilon^{-2}\log\left(\frac{2d}{\eta}\right)$ |
| Leverage score ($m_S$) | $(\rho+1)\log\left(\frac{4\rho}{\eta}\right)$ | $\kappa^{-1}\log\left(\frac{1}{\lambda\varepsilon}\right)\log\left(\frac{1}{\eta}\right)$ | $\left(\lambda^{-\frac{1}{2\beta}}\varepsilon^{-\frac{1}{\beta}}\right)^{2\wedge\frac{\beta}{\beta-1}}\log\left(\frac{1}{\eta}\right)$ |
| Lower bound on $\frac{m_O}{m_A}$ | $\varepsilon^{-2}\log\rho$ | $\varepsilon^{-2+h}\log 2d, \quad \forall h > 0$ | $\varepsilon^{1/\beta-2}\log(2d/\eta)$ |
| Lower bound on $\frac{m_S}{m_A}$ | $\log\rho$ | $\min\left(\log\left(\frac{1}{\eta}\right), \kappa^{-1}\log\left(\frac{1}{\lambda\varepsilon}\right)\right)$ | $\left(\lambda^{-\frac{1}{2\beta}}\varepsilon^{-\frac{1}{\beta}}\right)^{-1+2\wedge\frac{\beta}{\beta-1}}$ |

We illustrate numerically our predictions for adaptive Gaussian sketching versus oblivious Gaussian sketching. With $n = 1000$ and $d = 2000$, we generate matrices $A^{\text{exp}}$ and $A^{\text{poly}}$, with spectral decay satisfying respectively $\nu_j \sim ne^{-0.1j}$ and $\nu_j \sim nj^{-2}$. First, we perform binary logistic regression, with $f(Ax) = n^{-1}\sum_{i=1}^n \ell_{y_i}(a_i^\top x)$ where $\ell_{y_i}(z) = y_i\log(1+e^{-z}) + (1-y_i)\log(1+e^z)$, $y \in \{0,1\}^n$ and $a_i$ is the $i$-th row of $A$. For the polynomial (resp. exponential) decay, we expect the relative error $\|\widetilde{x} - x^*\|_2/\|x^*\|_2$ to follow w.h.p. a decay proportional to $m^{-1}$ (resp. $e^{-0.05m}$). Figure 1 confirms those predictions. We repeat the same experiments with a second loss function, $f(Ax) = (2n)^{-1}\sum_{i=1}^n (a_i^\top x)_+^2 - 2(a_i^\top x)y_i$. The latter is a convex relaxation of the penalty $\frac{1}{2}\|(Ax)_+ - y\|_2^2$ for fitting a shallow neural network with a ReLU non-linearity. Again, Figure 1 confirms our predictions, and we observe that the adaptive method performs much better than the oblivious sketch.

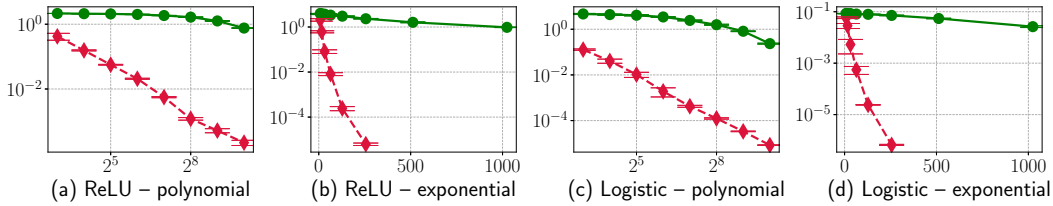

(a) ReLU – polynomial    (b) ReLU – exponential    (c) Logistic – polynomial    (d) Logistic – exponential

Figure 1: Relative error versus sketching dimension $m \in \{2^k \mid 3 \leqslant k \leqslant 10\}$ of adaptive Gaussian sketching (red) and oblivious Gaussian sketching (green), for the ReLU and logistic models, and the exponential and polynomial decays. We use $\lambda = 10^{-4}$ for all simulations. Results are averaged over 10 trials. Bar plots show (twice) the empirical standard deviations.

## 3 Algorithms for adaptive subspace sketching

### 3.1 Numerical conditioning and generic algorithm

A standard quantity to characterize the capacity of a convex program to be solved efficiently is its condition number [6], which, for the primal (1) and (adaptive) sketched program (2), are given by

$$\kappa = \frac{\lambda + \sup_x \sigma_1\left(A^\top\nabla^2 f(Ax)A\right)}{\lambda + \inf_x \sigma_d\left(A^\top\nabla^2 f(Ax)A\right)}, \quad \kappa_S = \frac{\sup_\alpha \sigma_1\left(\widetilde{S}^\top A(\lambda I + A^\top\nabla^2 f(AA^\top\widetilde{S}\alpha)A)A^\top\widetilde{S}\right)}{\inf_\alpha \sigma_d\left(\widetilde{S}^\top A(\lambda I + A^\top\nabla^2 f(AA^\top\widetilde{S}\alpha)A)A^\top\widetilde{S}\right)}.$$

The latter can be significantly larger than $\kappa$, up to $\kappa_S \approx \kappa\frac{\sigma_1(\widetilde{S}^\top AA^\top\widetilde{S})}{\sigma_m(\widetilde{S}^\top AA^\top\widetilde{S})} \gg \kappa$. A simple change of variable overcomes this issue. With $A_{S,\dagger} = AS(S^\top S)^{-\frac{1}{2}}$, we solve instead the optimization problem

$$\alpha_\dagger^* = \operatorname*{argmin}_{\alpha_\dagger \in \mathbb{R}^m} f(A_{S,\dagger}\alpha_\dagger) + \frac{\lambda}{2}\|\alpha_\dagger\|_2^2. \tag{7}$$

It holds that $\widetilde{x} = -\lambda^{-1}A^\top \nabla f(A_{S,\dagger}\alpha_\dagger^*)$. The additional complexity induced by this change of variables comes from computing the (square-root) pseudo-inverse of $S^\top S$, which requires $\mathcal{O}(m^3)$ flops via a singular value decomposition. When $m$ is small, this additional computation is negligible and numerically stable, and the re-scaled sketched program (7) is actually better conditioned that the original primal program (1), as stated in the next result that we prove in Appendix C.3.

**Proposition 3.** *Under adaptive sketching, the condition number $\kappa_\dagger$ of the re-scaled sketched program* (7) *satisfies $\kappa_\dagger \leqslant \kappa$ with probability* 1.

---

**Algorithm 1:** Generic algorithm for adaptive sketching.

---

**Input :** Data matrix $A \in \mathbb{R}^{n \times d}$, random matrix $\widetilde{S} \in \mathbb{R}^{n \times m}$ and parameter $\lambda > 0$.

1 Compute the sketching matrix $S = A^\top \widetilde{S}$, and, the sketched matrix $A_S = AS$.

2 Compute the re-scaling matrix $R = \left(S^\top S\right)^{-\frac{1}{2}}$, and the re-scaled sketched matrix $A_{S,\dagger} = A_S R$.

3 Solve the convex optimization problem (7), and return $\widetilde{x} = -\frac{1}{\lambda}A^\top \nabla f\left(A_{S,\dagger}\alpha_\dagger^*\right)$.

---

We observed a drastic practical performance improvement between solving the sketched program as formulated in (2) and its well-conditioned version (7).

If the chosen sketch dimension $m$ is itself prohibitively large for computing the matrix $Q = (S^\top S)^{-\frac{1}{2}}$, one might consider a pre-conditioning matrix $Q$, which is faster to compute, and such that the matrix $SQ$ is well-conditioned. Typically, one might compute a matrix $Q$ based on an approximate singular value decomposition of the matrix $S^\top S$. Then, one solves the optimization problem $\alpha_Q^* = \operatorname{argmin}_{\alpha \in \mathbb{R}^m} f(ASQ\alpha) + \frac{\lambda}{2}\|SQ\alpha\|_2^2$. Provided that $Q$ is invertible, it holds that $\widetilde{x}$ satisfies $\widetilde{x} = -\lambda^{-1}A^\top \nabla f(ASQ\alpha_Q^*)$.

## 3.2 Error contraction and almost exact recovery of the optimal solution

The estimate $\widetilde{x}$ satisfies a guarantee of the form $\|\widetilde{x}-x^*\|_2 \leqslant \varepsilon\|x^*\|_2$ w.h.p., and, with $\varepsilon < 1$ provided that $\lambda$ is large enough. Here, we extend Algorithm 1 to an iterative version which takes advantage of this error contraction, and which is relevant when a high-precision estimate $\widetilde{x}$ is needed.

---

**Algorithm 2:** Iterative adaptive sketching

---

**Input :** Data matrix $A \in \mathbb{R}^{n \times d}$, random matrix $\widetilde{S} \in \mathbb{R}^{n \times m}$, iterations number $T$, parameter $\lambda > 0$.

1 Compute the sketched matrix $A_{S,\dagger}$ as in Algorithm 1. Set $\widetilde{x}^{(0)} = 0$.

2 **for** $t = 1, 2, \ldots, T$ **do**

3     Compute $a^{(t)} = A\widetilde{x}^{(t-1)}$, and, $b^{(t)} = \left(S^\top S\right)^{-\frac{1}{2}} S^\top \widetilde{x}^{(t-1)}$.

4     Solve the following convex optimization problem

$$\alpha_\dagger^{(t)} = \operatorname*{argmin}_{\alpha_\dagger \in \mathbb{R}^m} f(A_{S,\dagger}\alpha_\dagger + a^{(t)}) + \frac{\lambda}{2}\|\alpha_\dagger + b^{(t)}\|_2^2. \tag{8}$$

    Update the solution by $\widetilde{x}^{(t)} = -\frac{1}{\lambda}A^\top \nabla f(A_{S,\dagger}\alpha_\dagger^{(t)} + a^{(t)})$.

5 **end**

6 Return the last iterate $\widetilde{x}^{(T)}$.

---

A key advantage is that, at each iteration, the same sketching matrix $S$ is used. Thus, the sketched matrix $A_{S,\dagger}$ has to be computed only once, at the beginning of the procedure. The output $\widetilde{x}^{(T)}$ satisfies the following recovery property, which empirical benefits are illustrated in Figure 2.

**Theorem 2.** *After $T$ iterations of Algorithm 2, provided that $\lambda \geqslant 2\mu Z_f^2$, it holds that*

$$\|\widetilde{x}^{(T)} - x^*\|_2 \leqslant \left(\frac{\mu Z_f^2}{2\lambda}\right)^{\frac{T}{2}}\|x^*\|_2. \tag{9}$$

*Further, if $S = A^\top \widetilde{S}$ where $\widetilde{S} \in \mathbb{R}^{n \times m}$ with i.i.d. Gaussian entries and $m = 2k$ for some target rank $k \geqslant 2$, then, for some universal constant $c_0 \leqslant 36$, after $T$ iterations of Algorithm 2, provided that $\lambda \geqslant 2c_0^2 \mu R_k^2(A)$, the approximate solution $\widetilde{x}^{(T)}$ satisfies with probability at least $1 - 12e^{-k}$,*

$$\|\widetilde{x}^{(T)} - x^*\|_2 \leqslant \left( \frac{c_0^2 \mu R_k^2(A)}{2\lambda} \right)^{\frac{T}{2}} \|x^*\|_2 \,. \tag{10}$$

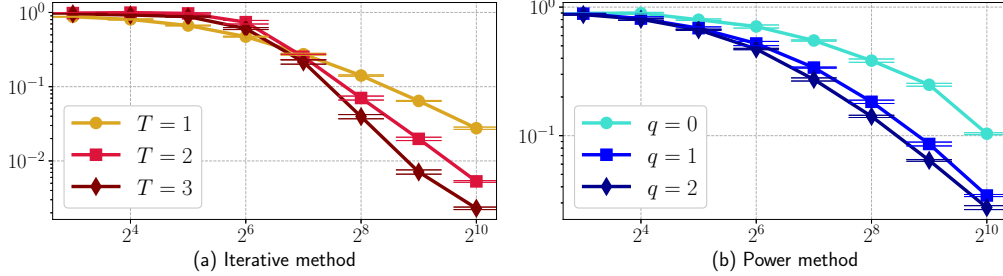

(a) Iterative method  (b) Power method

Figure 2: Relative error versus sketching dimension $m \in \{2^k \mid 3 \leqslant k \leqslant 10\}$ of adaptive Gaussian sketching for (a) the iterative method (Algorithm 2) and (b) the power method (see Remark 2). We use the MNIST dataset with images mapped through 10000-dimensional random Fourier features [20] for even-vs-odd classification using binary logistic loss, and, $\lambda = 10^{-5}$. Results are averaged over 20 trials. Bar plots show (twice) the empirical standard deviations.

**Remark 2.** *An immediate extension of Algorithms 1 and 2 consists in using the power method [15]. Given $q \in \mathbb{N}$, one uses the sketching matrix $S = (A^\top A)^q A^\top \widetilde{S}$. The larger $q$, the smaller the approximation error $\|AA^\top - AS(S^\top S)^\dagger S^\top A^\top\|_2$ (see Corollary 10.10 in [15]). Of pratical interest are data matrices $A$ with a spectral profile starting with a fast decay, and then becoming flat. This happens typically for $A$ of the form $A = \overline{A} + W$, where $\overline{A}$ has a fast decay and $W$ is a noise matrix with, for instance, independent subgaussian rows [26]. Our results easily extend to this setting and we illustrate its empirical benefits in Figure 2.*

## 4  Application to empirical risk minimization and kernel methods

By the representer theorem, the primal program (1) can be re-formulated as

$$w^* \in \underset{w \in \mathbb{R}^n}{\arg\min} f(Kw) + \frac{\lambda}{2} w^\top K w \,, \tag{11}$$

where $K = AA^\top$. Clearly, it holds that $x^* = A^\top w^*$. Given a matrix $\widetilde{S}$ with i.i.d. Gaussian entries, we consider the sketched version of the kernelized primal program (11),

$$\alpha^* \in \underset{\alpha \in \mathbb{R}^m}{\arg\min} f(K\widetilde{S}\alpha) + \frac{\lambda}{2} \alpha^\top \widetilde{S}^\top K \widetilde{S} \alpha \,. \tag{12}$$

The sketched program (12) is exactly our adaptive Gaussian sketched program (2). Thus, setting $\widetilde{w} = -\lambda^{-1} \nabla f(K\widetilde{S}\alpha^*)$, it holds that $\widetilde{x} = A^\top \widetilde{w}$. Since the relative error $\|\widetilde{x} - x^*\|_2 / \|x^*\|_2$ is controlled by the decay of the eigenvalues of $K$, so does the relative error $\|A^\top(\widetilde{w} - w^*)\|_2 / \|A^\top w^*\|_2$. More generally, the latter statements are still true if $K$ is any positive semi-definite matrix, and, if we replace $A^\top$ by any square-root matrix of $K$. Here, we denote $Z_f \equiv Z_f\left(K^{\frac{1}{2}}, K^{\frac{1}{2}}\widetilde{S}\right)$ (see Eq. (3)).

**Theorem 3.** *Let $K \in \mathbb{R}^{n \times n}$ be any positive semi-definite matrix. Let $w^*$ be any minimizer of the kernel program (11) and $\alpha^*$ be any minimizer of its sketched version (12). Define the approximate solution $\widetilde{w} = -\frac{1}{\lambda} \nabla f(K\widetilde{S}\alpha^*)$. If $\lambda \geqslant 2\mu Z_f^2$, then it holds that*

$$\|K^{\frac{1}{2}}(\widetilde{w} - w^*)\|_2 \leqslant \sqrt{\frac{\mu}{2\lambda}} Z_f \|K^{\frac{1}{2}} w^*\|_2 \,. \tag{13}$$

For a positive definite kernel $k : \mathbb{R}^d \times \mathbb{R}^d \to \mathbb{R}$ and a data matrix $A = [a_1, \ldots, a_n]^\top \in \mathbb{R}^{n \times d}$, let $K$ be the empirical kernel matrix, with $K_{ij} = k(a_i, a_j)$. Let $\varphi(\cdot) \in \mathbb{R}^D$ be a random feature map [20, 8], such as random Fourier features or a random convolutional neural net. We are interested in the computational complexities of forming the sketched versions of the primal (1), the kernel primal (11) and the primal (1) with $\varphi(A)$ instead of $A$. We compare the complexities of adaptive and oblivious sketching and uniform column sub-sampling. Table 2 shows that all three methods have similar complexities for computing $AS$ and $\varphi(A)S$. Adaptive sketching exhibits an additional factor 2 that comes from computing the correlated sketching matrices $S = A^\top \widetilde{S}$ and $S = \varphi(A)^\top \widetilde{S}$. In practice, the latter is negligible compared to the cost of forming $\varphi(A)$ which, for instance, corresponds to a forward pass over the whole dataset in the case of a convolutional neural network. On the other hand, uniform column sub-sampling is significantly faster in order to form the sketched kernel matrix $K\widetilde{S}$, which relates to the well-known computational advantages of kernel Nystrom methods [30].

Table 2: Complexity of forming the sketched programs, given $A \in \mathbb{R}^{n \times d}$. We denote $d_k$ the number of flops to evaluate the kernel product $k(a, a')$, and, $d_\varphi$ the number of flops for a forward-pass $\varphi(a)$. Note that these complexities could be reduced through parallelization.

| | $AS$ | $\varphi(A)S$ | $K\widetilde{S}$ |
|---|---|---|---|
| Adaptive sketching | $\mathcal{O}(2mdn)$ | $\mathcal{O}(d_\varphi n) + \mathcal{O}(2mDn)$ | $\mathcal{O}(d_k n^2) + \mathcal{O}(mn^2)$ |
| Oblivious sketching | $\mathcal{O}(mdn)$ | $\mathcal{O}(d_\varphi n) + \mathcal{O}(mDn)$ | - |
| Uniform column sub-sampling | $\mathcal{O}(mdn)$ | $\mathcal{O}(d_\varphi n) + \mathcal{O}(mDn)$ | $\mathcal{O}(d_k nm)$ |

## 5 Numerical evaluation of adaptive Gaussian sketching

We evaluate Algorithm 1 on MNIST and CIFAR10. First, we aim to show that the sketching dimension can be considerably smaller than the original dimension while retaining (almost) the same test classification accuracy. Second, we aim to get significant speed-ups in achieving a high-accuracy classifier. To solve the primal program (1), we use two standard algorithms, stochastic gradient descent (SGD) with (best) fixed step size and stochastic variance reduction gradient (SVRG) [16] with (best) fixed step size and frequency update of the gradient correction. To solve the adaptive sketched program (2), we use SGD, SVRG and the sub-sampled Newton method [4, 10] – which we refer to as Sketch-SGD, Sketch-SVRG and Sketch-Newton. The latter is well-suited to the sketched program, as the low-dimensional Hessian matrix can be quickly inverted at each iteration. For both datasets, we use 50000 training and 10000 testing images. We transform each image using a random Fourier feature map $\varphi(\cdot) \in \mathbb{R}^D$, i.e., $\langle \varphi(a), \varphi(a') \rangle \approx \exp\left(-\gamma \|a - a'\|_2^2\right)$ [20, 18]. For MNIST and CIFAR10, we choose respectively $D = 10000$ and $\gamma = 0.02$, and, $D = 60000$ and $\gamma = 0.002$, so that the primal is respectively 10000-dimensional and 60000-dimensional. Then, we train a classifier via a sequence of binary logistic regressions – which allow for efficient computation of the Hessian and implementation of the Sketch-Newton algorithm –, using a one-vs-all procedure.

First, we evaluate the test classification error of $\widetilde{x}$. We solve to optimality the primal and sketched programs for values of $\lambda \in \{10^{-4}, 5 \cdot 10^{-5}, 10^{-5}, 5 \cdot 10^{-6}\}$ and sketching dimensions $m \in \{64, 128, 256, 512, 1024\}$. In Table 3 are reported the results, which are averaged over 20 trials for MNIST and 10 trials for CIFAR10, and, empirical variances are reported in Appendix A. Overall, the adaptive sketched program yields a high-accuracy classifier for most couples $(\lambda, m)$. Further, we match the best primal classifier with values of $m$ as small as 256 for MNIST and 512 for CIFAR10, which respectively corresponds to a dimension reduction by a factor $\approx 40$ and $\approx 120$. These results additionally suggest that adaptive Gaussian sketching introduces an implicit regularization effect, which might be related to the benefits of spectral cutoff estimators. For instance, on CIFAR10, using $\lambda = 10^{-5}$ and $m = 512$, we obtain an improvement in test accuracy by more than 2% compared to $x^*$. Further, over some sketching dimension threshold under which the performance is bad, as the value of $m$ increases, the test classification error of $\widetilde{x}$ increases to that of $x^*$, until matching it.

Further, we evaluate the test classification error of two sketching baselines, that is, oblivious Gaussian sketching for which the matrix $S$ has i.i.d. Gaussian entries, and, adaptive column sub-sampling (Nystrom method) for which $S = A^\top \widetilde{S}$ with $\widetilde{S}$ a column sub-sampling matrix. As reported in Table 4, adaptive Gaussian sketching performs better for a wide range of values of sketching size $m$ and regularization parameter $\lambda$.

Table 3: Test classification error of adaptive Gaussian sketching on MNIST and CIFAR10 datasets.

| $\lambda$ | $x^*_{\text{MNIST}}$ | $\widetilde{x}_{64}$ | $\widetilde{x}_{128}$ | $\widetilde{x}_{256}$ | $\widetilde{x}_{512}$ | $\widetilde{x}_{1024}$ | $x^*_{\text{CIFAR}}$ | $\widetilde{x}_{64}$ | $\widetilde{x}_{128}$ | $\widetilde{x}_{256}$ | $\widetilde{x}_{512}$ | $\widetilde{x}_{1024}$ |
|---|---|---|---|---|---|---|---|---|---|---|---|---|
| $10^{-4}$ | 5.4 | 4.8 | 4.8 | 5.2 | 5.3 | 5.4 | - | - | - | - | - | - |
| $5 \cdot 10^{-5}$ | 4.6 | 4.1 | 3.8 | 4.0 | 4.3 | 4.5 | 51.6 | 52.1 | 50.5 | 50.6 | 50.8 | 51.0 |
| $10^{-5}$ | 2.8 | 8.1 | 3.4 | 2.4 | 2.5 | 2.8 | 48.2 | 60.1 | 54.5 | 47.7 | 45.9 | 46.2 |
| $5 \cdot 10^{-6}$ | 2.5 | 11.8 | 4.9 | 2.8 | 2.6 | 2.4 | 47.6 | 63.6 | 59.8 | 51.9 | 47.7 | 45.8 |

Table 4: Test classification error on MNIST and CIFAR10. "AG": Adaptive Gaussian sketch, "Ob": Oblivious Gaussian sketch, "N": Nystrom method.

| $\lambda$ | $x^*_{\text{MNIST}}$ | $\widetilde{x}^{AG}_{256}$ | $\widetilde{x}^{AG}_{1024}$ | $\widetilde{x}^{Ob}_{256}$ | $\widetilde{x}^{Ob}_{1024}$ | $\widetilde{x}^{N}_{256}$ | $\widetilde{x}^{N}_{1024}$ |
|---|---|---|---|---|---|---|---|
| $5 \cdot 10^{-5}$ | 4.6 % | 4.0 % | 4.5 % | 25.2 % | 8.5 % | 5.0 % | 4.6 |
| $5 \cdot 10^{-6}$ | 2.5% | 2.8% | 2.4% | 30.1% | 9.4% | 3.0% | 2.7% |

| $\lambda$ | $x^*_{\text{CIFAR}}$ | $\widetilde{x}^{AG}_{256}$ | $\widetilde{x}^{AG}_{1024}$ | $\widetilde{x}^{Ob}_{256}$ | $\widetilde{x}^{Ob}_{1024}$ | $\widetilde{x}^{N}_{256}$ | $\widetilde{x}^{N}_{1024}$ |
|---|---|---|---|---|---|---|---|
| $5 \cdot 10^{-5}$ | 51.6 % | 50.6% | 51.0% | 88.2% | 70.5% | 55.8% | 53.1% |
| $5 \cdot 10^{-6}$ | 47.6% | 51.9% | 45.8% | 88.9% | 80.1% | 57.2% | 55.8% |

Then, we compare the test classification error versus wall-clock time of the optimization algorithms mentioned above. Figure 3 shows results for some values of $m$ and $\lambda$. We observe some speed-ups on the 10000-dimensional MNIST problem, in particular for Sketch-SGD and for Sketch-SVRG, for which computing the gradient correction is relatively fast. Such speed-ups are even more significant on the 60000-dimensional CIFAR10 problem, especially for Sketch-Newton. A few iterations of Sketch-Newton suffice to almost reach the minimum $\widetilde{x}$, with a per-iteration time which is relatively small thanks to dimensionality reduction. Hence, it is more than 10 times faster to reach the best test accuracy using the sketched program. In addition to random Fourier features mapping, we carry out another set of experiments with the CIFAR10 dataset, in which we pre-process the images. That is, similarly to [24, 21], we map each image through a random convolutional layer. Then, we kernelize these processed images using a Gaussian kernel with $\gamma = 2 \cdot 10^{-5}$. Using our implementation, the best test accuracy of the kernel primal program (11) we obtained is 73.1%. Sketch-SGD, Sketch-SVRG and Sketch-Newton – applied to the sketched kernel program (12) – match this test accuracy, with significant speed-ups, as reported in Figure 3.

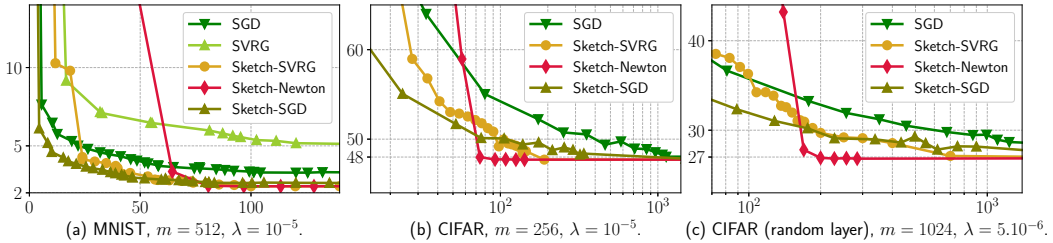

(a) MNIST, $m = 512$, $\lambda = 10^{-5}$.  (b) CIFAR, $m = 256$, $\lambda = 10^{-5}$.  (c) CIFAR (random layer), $m = 1024$, $\lambda = 5.10^{-6}$.

Figure 3: Test classification error (percentage) versus wall-clock time (seconds).

## Acknowledgements

This work was partially supported by the National Science Foundation under grant IIS-1838179 and Office of Naval Research, ONR YIP Program, under contract N00014-17-1-2433.

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
