[Supplementary Material · camera_ready_supp.pdf]

# A  Additional experimental results and implementation details

## A.1  Synthetic examples (Figure 1)

With $n = 1000$ and $d = 2000$, we sample two matrices with orthonormal columns $U \in \mathbb{R}^{n \times n}$ and $V \in \mathbb{R}^{d \times n}$, uniformly at random from their respective spaces. We construct two diagonal matrices $\Sigma^{\text{poly}}, \Sigma^{\text{exp}} \in \mathbb{R}^{n \times n}$, such that their respective diagonal elements are $\Sigma^{\text{poly}}_{jj} = \sqrt{n}j^{-1}$ and $\Sigma^{\text{exp}}_{jj} = \sqrt{n}e^{-0.05j}$. We set $A^{\text{exp}} = U\Sigma^{\text{exp}}V^\top$ and $A^{\text{poly}} = U\Sigma^{\text{poly}}V^\top$, and we sample a planted vector $x_{\text{gd}} \in \mathbb{R}^d$ with iid entries $\mathcal{N}(0, 1)$.

In the case of binary logistic regression, for each $A \in \{A^{\text{exp}}, A^{\text{poly}}\}$, we set $y_i = 0.5\left(\text{sign}\left(\langle a_i, x_{\text{gd}}\rangle\right) + 1\right)$, for $i = 1, \dots, n$.

For the ReLU model, we set $y_i = (\langle a_i, x_{\text{gd}}\rangle)_+$, where $(z)_+ = \max(0, z)$. Hence, each observation $y_i$ is the result of a linear operation $z_i = \langle x, a_i \rangle$ and a non-linear operation $y_i = (z_i)_+$. Additionally, it can be shown that the global minimum of the optimization problem

$$\min_x \frac{1}{2} \sum_{i=1}^n (a_i^\top x)_+^2 - 2(a_i^\top x)y_i,$$

is equal to $x_{\text{gd}}$, which motivates using such a convex relaxation.

## A.2  Numerical illustration of the iterative and power methods (Figure 2)

We use the MNIST dataset with 50000 training images and 10000 testing images. We rescale the pixel values between $[0, 1]$. Each image is mapped through random cosines $\varphi(\cdot) \in \mathbb{R}^D$ which approximate the Gaussian kernel, i.e., $\langle \varphi(a), \varphi(a')\rangle \approx \exp(-\gamma\|a - a^2\|_2^2)$. We choose $D = 10000$ and $\gamma = 0.02$.

We perform binary logistic regression for even-vs-odd classification of the digits.

For the iterative method, we use the sketching matrix $S = (A^\top A)^2 A^\top \widetilde{S}$, where $\widetilde{S}$ is Gaussian iid. That is, we run the iterative method on top of the power method, with $q = 2$.

## A.3  Adaptive Gaussian sketching on MNIST and CIFAR10 datasets (Table 3 and Figure 3)

Table 5: Empirical standard deviation of test classification error on MNIST and CIFAR10 datasets, mapped through Gaussian random Fourier features, respectively with $D = 10000$ and $\gamma = 0.02$, and, $D = 60000$ and $\gamma = 0.002$. The notation $\widetilde{x}_m$ refers to the solution of (2), with sketching size $m$.

| $\lambda$ | $x^*_{\text{MNIST}}$ | $\widetilde{x}_{64}$ | $\widetilde{x}_{128}$ | $\widetilde{x}_{256}$ | $\widetilde{x}_{512}$ | $\widetilde{x}_{1024}$ | $x^*_{\text{CIFAR}}$ | $\widetilde{x}_{64}$ | $\widetilde{x}_{128}$ | $\widetilde{x}_{256}$ | $\widetilde{x}_{512}$ | $\widetilde{x}_{1024}$ |
|---|---|---|---|---|---|---|---|---|---|---|---|---|
| $10^{-4}$ | - | 0.2 | 0.2 | 0.1 | 0.1 | 0.1 | - | - | - | - | - | - |
| $5 \cdot 10^{-5}$ | - | 0.2 | 0.2 | 0.2 | 0.1 | 0.1 | - | 0.5 | 0.3 | 0.3 | 0.2 | 0.2 |
| $10^{-5}$ | - | 2.0 | 0.8 | 0.2 | 0.1 | 0.1 | - | 4.8 | 3.2 | 0.6 | 0.2 | 0.2 |
| $5 \cdot 10^{-6}$ | - | 3.2 | 2.1 | 0.3 | 0.2 | 0.1 | - | 4.1 | 3.5 | 2.1 | 0.6 | 0.6 |

Experiments were run in Python on a workstation with 20 cores and 256 GB of RAM. The MNIST and CIFAR10 datasets were downloaded through the PyTorch.torchvision module and converted to NumPy arrays. We use the Sklearn.kernel_approximation.RBFSampler module to generate random cosines. We use our own implementation of each algorithm for a fair comparison.

For SGD, we use a batch size equal to 128. For SVRG, we use a batch size equal to 128 and update the gradient correction every 400 iterations. For Sketch-SGD, we use a batch size equal to 1024. For Sketch-SVRG, we use a batch size equal to 64 and update the gradient correction every 200 iterations. Each iteration of the sub-sampled Newton method (Sketch-Newton) computes a full-batch gradient, and, the Hessian with respect to a batch of size 1500.

For SGD and SVRG, we considered step sizes $\eta$ between $10^{-2}$ and $10^2$. We obtained best performance for $\eta = 10^1$. For the sub-sampled Newton method, we use a step size $\eta = 1$, except for the first 5 iterations, for which we use $\eta = 0.2$.

In Figure 3, we did not report results for SVRG for solving the primal (1) on CIFAR, as the computation time for reaching a satisfying performance was significantly larger than for the other algorithms.

In Table 3, we did not investigate results for CIFAR with $\lambda = 10^{-4}$, as the primal classifier had a test error significantly larger than smaller values of $\lambda$.

## B  Proofs of main results

Here, we establish our main technical results, that is, the deterministic regret bounds (4) and (5) stated in Theorem 1 and its high-probability version stated in Corollary 1, along with its extension to the iterative Algorithm 2 as given in Theorem 2, and its variant for kernel methods, given in Theorem 3. Our analysis is based on convex analysis and Fenchel duality arguments.

### B.1  Proof of Theorem 1

We introduce the Fenchel dual program of (1),

$$\min_z f^*(z) + \frac{1}{2\lambda} \|A^\top z\|_2^2. \tag{14}$$

For a sketching matrix $S \in \mathbb{R}^{d \times m}$, the Fenchel dual program of (2) is

$$\min_y f^*(y) + \frac{1}{2\lambda} \|P_S A^\top y\|_2^2. \tag{15}$$

Let $\alpha^*$ be any minimizer of the sketched program (2). Then, according to Proposition 2, the unique solution of the dual sketched program (15) is

$$y^* = \nabla f(AS\alpha^*)$$

and the subgradient set $\partial f^*(y^*)$ is non-empty. We fix $g_{y^*} \in \partial f^*(y^*)$.

According to Proposition 1, the dual program (14) admits a unique solution $z^*$, which satisfies

$$z^* = \nabla f(Ax^*),$$

and which subgradient set $\partial f^*(z^*)$ is non-empty. We fix $g_{z^*} \in \partial f^*(z^*)$.

We denote the error between the two dual solutions by $\Delta = y^* - z^*$. By optimality of $y^*$ with respect to the sketched dual (15) and by feasibility of $z^*$, first-order optimality conditions imply that

$$\langle \frac{1}{\lambda} A P_S A^\top y^* + g_{y^*}, \Delta \rangle \leqslant 0.$$

Similarly, by optimality of $z^*$ with respect to the dual (14) and by feasibility of $y^*$, we get by first-order optimality conditions that

$$\langle \frac{1}{\lambda} A A^\top z^* + g_{z^*}, \Delta \rangle \geqslant 0.$$

It follows that

$$
\begin{aligned}
\langle \frac{1}{\lambda} A P_S A^\top \Delta, \Delta \rangle &= \langle \frac{1}{\lambda} A P_S A^\top y^*, \Delta \rangle - \langle \frac{1}{\lambda} A P_S A^\top z^*, \Delta \rangle \\
&= \underbrace{\langle \frac{1}{\lambda} A P_S A^\top y^* + g_{y^*}, \Delta \rangle}_{\leqslant 0} + \langle g_{z^*} - g_{y^*}, \Delta \rangle - \langle \frac{1}{\lambda} A P_S A^\top z^* + g_{z^*}, \Delta \rangle \\
&\leqslant \langle g_{z^*} - g_{y^*}, \Delta \rangle - \langle \frac{1}{\lambda} A P_S A^\top z^* + g_{z^*}, \Delta \rangle \\
&= \langle g_{z^*} - g_{y^*}, \Delta \rangle + \langle \frac{1}{\lambda} A P_S^\perp A^\top z^*, \Delta \rangle - \underbrace{\langle \frac{1}{\lambda} A A^\top z^* + g_{z^*}, \Delta \rangle}_{\geqslant 0} \\
&\leqslant \langle g_{z^*} - g_{y^*}, \Delta \rangle + \langle \frac{1}{\lambda} A P_S^\perp A^\top z^*, \Delta \rangle.
\end{aligned}
\tag{16}
$$

Strong $\mu$-smoothness of $f$ implies that the function $f^*$ is $\frac{1}{\mu}$-strongly convex. Hence, it follows that

$$\langle g_{z^*} - g_{y^*}, \Delta \rangle + \frac{1}{\mu} \|\Delta\|_2^2 \leqslant 0. \tag{17}$$

Therefore, combining (17) with the previous set of inequalities (16), we get

$$\langle \frac{1}{\lambda} A P_S A^\top \Delta, \Delta \rangle + \frac{1}{\mu} \|\Delta\|_2^2 \leqslant \langle \frac{1}{\lambda} A P_S^\perp A^\top z^*, \Delta \rangle,$$

and, multiplying both sides by $\lambda$,

$$\langle A P_S A^\top \Delta, \Delta \rangle + \frac{\lambda}{\mu} \|\Delta\|_2^2 \leqslant \langle A P_S^\perp A^\top z^*, \Delta \rangle. \tag{18}$$

By definition of $Z_f$ and since $\Delta \in \mathrm{dom} f^* - z^*$, it holds that

$$\frac{\Delta^\top A P_S^\perp A^\top \Delta}{\|\Delta\|_2^2} \leqslant Z_f^2,$$

which we can rewrite as

$$\langle A P_S A^\top \Delta, \Delta \rangle \geqslant \langle A A^\top \Delta, \Delta \rangle - Z_f^2 \|\Delta\|_2^2. \tag{19}$$

Hence, combining (19) and (18), we obtain

$$\left( \frac{\lambda}{\mu} - Z_f^2 \right) \|\Delta\|_2^2 + \|A^\top \Delta\|_2^2 \leqslant \langle A P_S^\perp A^\top z^*, \Delta \rangle, \tag{20}$$

Under the assumption that $\lambda \geqslant 2\mu Z_f^2$, it holds that $\lambda/\mu - Z_f^2 \geqslant \lambda/(2\mu)$. Thus,

$$\left( \frac{\lambda}{\mu} - Z_f^2 \right) \|\Delta\|_2^2 + \|A^\top \Delta\|_2^2 \geqslant \frac{\lambda}{2\mu} \|\Delta\|_2^2 + \|A^\top \Delta\|_2^2$$

$$\geqslant \sqrt{\frac{2\lambda}{\mu}} \|\Delta\|_2 \|A^\top \Delta\|_2.$$

where we used the fact that for any $a, b \geqslant 0$, $a + b \geqslant 2\sqrt{ab}$, with $a = \frac{\lambda}{2\mu} \|\Delta\|_2^2$ and $b = \|A^\top \Delta\|_2^2$. Combining the former inequality with inequality (20), we obtain

$$\sqrt{\frac{2\lambda}{\mu}} \|\Delta\|_2 \|A^\top \Delta\|_2 \leqslant \langle A P_S^\perp A^\top z^*, \Delta \rangle. \tag{21}$$

The right-hand side of the latter inequality can be bounded as

$$\langle A P_S^\perp A^\top z^*, \Delta \rangle = \langle A^\top z^*, P_S^\perp A^\top \Delta \rangle$$

$$\underset{(i)}{\leqslant} \|A^\top z^*\|_2 \|P_S^\perp A^\top \Delta\|_2$$

$$\leqslant \|A^\top z^*\|_2 \|\Delta\|_2 \sup_{\Delta \in \mathrm{dom} f^* - z^*} \left( \frac{\|P_S^\perp A^\top \Delta\|_2}{\|\Delta\|_2} \right)$$

$$\underset{(ii)}{=} \|A^\top z^*\|_2 \|\Delta\|_2 Z_f.$$

where $(i)$ follows from Cauchy-Schwartz inequality, and $(ii)$ holds by definition of $Z_f$. Thus, inequality (21) becomes

$$\sqrt{\frac{2\lambda}{\mu}} \|\Delta\|_2 \|A^\top \Delta\|_2 \leqslant \|A^\top z^*\|_2 \|\Delta\|_2 Z_f. \tag{22}$$

From Propositions 1 and 2, we have that $A^\top z^* = -\lambda x^*$ and $y^* = \nabla f(AS\alpha^*)$. By definition of $\widetilde{x}$, it follows that $A^\top \Delta = -\lambda (\widetilde{x} - x^*)$. Then, rearranging inequality (22),

$$\|\widetilde{x} - x^*\|_2 \leqslant \sqrt{\frac{\mu}{2\lambda}} Z_f \|x^*\|_2,$$

which is exactly the desired regret bound (4). The regret bound (5) immediately follows from the fact that $Z_f \leqslant \|P_S^\perp A^\top\|_2$.

## B.2  Proof of Corollary 1

The proof combines our deterministic regret bound (5), along with the following result, which is a re-writing of Corollary 10.9, in [15].

**Lemma 1.** *Let $k \geqslant 2$ be a target rank and $m \geqslant 1$ a sketching dimension such that $k < m \leqslant \min(n, d)$. Let $\widetilde{S}$ be an $n \times m$ random matrix with iid Gaussian entries. Define the oversampling ratio $r = (m - k)/k$, and the sketching matrix $S = A^\top \widetilde{S}$. Then, provided $rk \geqslant 4$, it holds with probability at least $1 - 6e^{-rk}$ that*

$$\|P_S^\perp A^\top\|_2 \leqslant \frac{c_0}{\sqrt{2}} \sqrt{\frac{r+1}{r}} \left( \sigma_{k+1}^2 + \frac{1}{rk} \sum_{j=k+1}^{\min(n,d)} \sigma_j^2 \right)^{\frac{1}{2}}. \tag{23}$$

*where $\sigma_1 \geqslant \sigma_2 \geqslant \ldots$ are the singular values of $A$, and the universal constant $c_0$ satisfies $c_0 \leqslant 36$. In particular, if $m = 2k$, then it holds with probability at least $1 - 6e^{-k}$ that*

$$\|P_S^\perp A^\top\|_2 \leqslant c_0 \left( \sigma_{k+1}^2 + \frac{1}{k} \sum_{j=k+1}^{\min(n,d)} \sigma_j^2 \right)^{\frac{1}{2}} \tag{24}$$

$$= c_0 R_k(A).$$

From Theorem 1, if $\lambda \geqslant 2\mu Z_f^2$, then

$$\|\widetilde{x} - x^*\|_2 \leqslant \sqrt{\frac{\mu}{2\lambda}} \|P_S^\perp A^\top\|_2 \|x^*\|_2.$$

Hence, combining the latter inequality with Lemma 1, provided $2 \leqslant k \leqslant \frac{1}{2} \min(n, d)$, $m = 2k$ and $\lambda \geqslant 2\mu Z_f^2$, it holds with probability at least $1 - 6e^{-k}$ that

$$\|\widetilde{x} - x^*\|_2 \leqslant c_0 \sqrt{\frac{\mu}{2\lambda}} R_k(A) \|x^*\|_2.$$

We want to establish the latter inequality, but under the condition $\lambda \geqslant 2c_0^2 \mu R_k^2(A)$. But, by Lemma 1, the condition $\lambda \geqslant 2c_0^2 \mu R_k^2(A)$ implies that $\lambda \geqslant 2\mu Z_f^2$ with probability at least $1 - 6e^{-k}$. By union bound, it follows that if $\lambda \geqslant 2c_0^2 \mu R_k^2(A)$, then

$$\|\widetilde{x} - x^*\|_2 \leqslant c_0 \sqrt{\frac{\mu}{2\lambda}} R_k(A) \|x^*\|_2,$$

with probability at least $1 - 12e^{-k}$.

## B.3  Proof of Theorem 2

First, we show that for any $t \geqslant 0$, provided $\lambda \geqslant 2\mu Z_f^2$,

$$\|\widetilde{x}^{(t+1)} - x^*\|_2 \leqslant \sqrt{\frac{\mu}{2\lambda}} Z_f \|\widetilde{x}^{(t)} - x^*\|_2. \tag{25}$$

It should be noted that for $t = 0$, the latter inequality is exactly the regret bound (4). The proof for $t > 0$ follows similar steps.

Fix $t \geqslant 0$. Consider the optimization problem

$$\min_{\delta \in \mathbb{R}^d} f(A\delta + A\widetilde{x}^{(t)}) + \frac{\lambda}{2} \|\delta + \widetilde{x}^{(t)}\|_2^2. \tag{26}$$

which is equivalent to the primal program (1), up to a translation of the optimization variable. Thus, the unique optimal solution of (26) – which exists by strong convexity of the objective – is given by $\delta^* = x^* - \widetilde{x}^{(t)}$. By Fenchel duality (Corollary 31.2.1, [22]), it holds that

$$\min_{\delta} f(A\delta + A\widetilde{x}^{(t)}) + \frac{\lambda}{2} \|\delta + \widetilde{x}^{(t)}\|_2^2 = \max_{z} -f^*(z) - \frac{1}{2\lambda} z^\top A A^\top z,$$

and the optimal dual solution $z^*$ exists and is unique (by strong concavity of the dual objective). Further, by the Karush-Kuhn-Tucker conditions (Theorem 31.3, [22]), we have

$$\begin{cases} \delta^* = -\widetilde{x}^{(t)} - \frac{1}{\lambda}A^\top z^* \\ z^* = \nabla f(A\delta^* + A\widetilde{x}^{(t)}). \end{cases}$$

Observe that, by using the change of variables $\alpha = \left(S^\top S\right)^{-\frac{1}{2}}\alpha_\dagger$, the optimization problem (8) can be rewritten as

$$\min_{\alpha \in \mathbb{R}^m} f(AS\alpha + A\widetilde{x}^{(t)}) + \lambda\alpha^\top S^\top \widetilde{x}^{(t)} + \frac{\lambda}{2}\|S\alpha\|_2^2$$

$$\equiv \min_{\alpha \in \mathbb{R}^m} f(AS\alpha + A\widetilde{x}^{(t)}) + \frac{\lambda}{2}\|S\alpha + \widetilde{x}^{(t)}\|_2^2.$$

Let $\alpha_\dagger^{(t+1)}$ be the unique solution of (8). Then, setting $\alpha^{(t+1)} = \left(S^\top S\right)^{-\frac{1}{2}}\alpha_\dagger^{(t+1)}$, we have

$$\alpha^{(t+1)} \in \operatorname*{argmin}_{\alpha \in \mathbb{R}^m} f(AS\alpha + A\widetilde{x}^{(t)}) + \frac{\lambda}{2}\|S\alpha + \widetilde{x}^{(t)}\|_2^2. \tag{27}$$

By Fenchel duality, we get

$$\min_\alpha f(AS\alpha + A\widetilde{x}^{(t)}) + \frac{\lambda}{2}\|S\alpha + \widetilde{x}^{(t)}\|_2^2 = \max_y -f^*(y) + y^\top A\widetilde{x}^{(t)} - \frac{1}{2\lambda}y^\top AP_S A^\top y + \frac{\lambda}{2}\widetilde{x}^{(t)}P_S^\perp \widetilde{x}^{(t)}.$$

By strong concavity of the dual objective, there exists a unique maximizer $y^*$. Further, by the Karush-Kuhn-Tucker conditions (Theorem 31.3, [22]), we have

$$AS\alpha^{(t+1)} + A\widetilde{x}^{(t)} \in \partial f^*(y^*)$$

and, thus, $y^* = \nabla f\left(AS\alpha^{(t+1)} + A\widetilde{x}^{(t)}\right)$.

We define $\Delta = y^* - z^*$. Following similar steps as in the proof of Theorem 1, we obtain

$$\|A^\top \Delta\|_2^2 + \left(\frac{\lambda}{\mu} - Z_f^2\right)\|\Delta\|_2^2 \leqslant \langle AP_S^\perp(\lambda\widetilde{x}^{(t)} + A^\top z^*), \Delta\rangle$$

$$= -\lambda\langle AP_S^\perp \delta^*, \Delta\rangle.$$

Since $\lambda \geqslant 2\mu Z_f^2$, it follows that $\lambda/\mu - Z_f^2 \geqslant \lambda/(2\mu)$. Using the fact that for any $a, b \geqslant 0$, we have $2\sqrt{ab} \leqslant a + b$, we obtain the inequality

$$\sqrt{\frac{2\lambda}{\mu}}\|A^\top \Delta\|_2\|\Delta\|_2 \leqslant -\lambda\langle AP_S^\perp \delta^*, \Delta\rangle$$

$$\leqslant \lambda\|\delta^*\|_2 Z_f\|\Delta\|_2.$$

Dividing both sides by $\lambda\|\Delta\|_2$ and using the identities $\delta^* = x^* - \widetilde{x}^{(t)}$ and $A^\top \Delta/\lambda = x^* - \widetilde{x}^{(t+1)}$, we obtain the desired contraction inequality

$$\|\widetilde{x}^{(t+1)} - x^*\|_2 \leqslant \sqrt{\frac{\mu}{2\lambda}}Z_f\|\widetilde{x}^{(t)} - x^*\|_2.$$

By induction, it immediately follows that for any number of iterations $T \geqslant 1$,

$$\|\widetilde{x}^{(T)} - x^*\|_2 \leqslant \left(\frac{\mu}{2\lambda}Z_f^2\right)^{\frac{T}{2}}\|x^*\|_2.$$

The high-probability version follows by immediate application of Lemma 1 to the previous inequality.

### B.4   Proof of Theorem 3

Define

$$x^* = \operatorname*{argmin}_x f(K^{\frac{1}{2}}x) + \frac{\lambda}{2}\|x\|_2^2.$$

Set $\widetilde{x} = -\frac{1}{\lambda}K^{\frac{1}{2}}\nabla f(K\widetilde{S}\alpha^*)$, and

$$Z_f = Z_f\left(K^{\frac{1}{2}}, K^{\frac{1}{2}}\widetilde{S}\right).$$

Then, by application of Theorem 1 with $S = K^{\frac{1}{2}}\widetilde{S}$, it holds that $\|\widetilde{x} - x^*\|_2 \leqslant \sqrt{\mu/(2\lambda)}Z_f\|x^*\|_2$, provided that $\lambda \geqslant 2\mu Z_f^2$. We conclude by using the facts that $\widetilde{x} = K^{\frac{1}{2}}\widetilde{w}$ and $x^* = K^{\frac{1}{2}}w^*$.

# C Proofs of intermediate results

## C.1 Proposition 1 – Strong duality and Karush-Kuhn-Tucker conditions of the primal objective (1)

Denote $g(x) = \frac{\lambda}{2}\|x\|_2^2$. The functions $f$ and $g$ are proper, closed, convex and their domains are respectively equal to $\mathbb{R}^n$ and $\mathbb{R}^d$. It is then trivial that for any $x \in \mathbb{R}^d$, we have $x \in \text{dom}(g)$ and $Ax \in \text{dom}(f)$. Hence, all conditions to apply strong Fenchel duality results hold (Theorem 31.2, [22]). Using the fact that $g^*(z) = \frac{1}{2\lambda}\|z\|_2^2$, we get

$$\inf_x f(Ax) + \frac{\lambda}{2}\|x\|_2^2 = \sup_z -f^*(z) - \frac{1}{2\lambda}\|A^\top z\|_2^2,$$

and the supremum is attained for some $z^* \in \text{dom}(f^*)$. The uniqueness of $z^*$ follows from strong concavity of the dual objective, which comes from the $\frac{1}{\mu}$-strong convexity of $f^*$.

Further, the primal objective is also strongly convex over $\mathbb{R}^d$, which implies the existence and uniqueness of a minimizer $x^*$.

The Karush-Kuhn-Tucker conditions (Theorem 31.3, [22]) imply that $Ax^* \in \partial f^*(z^*)$. Since $\partial f^* = (\nabla f)^{-1}$ (Theorem 23.5, [22]), it follows that $z^* = \nabla f(Ax^*)$. Finally, by first-order optimality conditions of $x^*$, we have that $A^\top \nabla f(Ax^*) + \lambda x^* = 0$, i.e., $x^* = -\lambda^{-1}A^\top z^*$.

## C.2 Proposition 2 – Strong duality and Karush-Kuhn-Tucker conditions of the sketched primal objective (1)

Denote $g(\alpha) = \frac{\lambda}{2}\|S\alpha\|_2^2$. The functions $f$ and $g$ are proper, closed, convex and their domains are respectively equal to $\mathbb{R}^n$ and $\mathbb{R}^m$. It is then trivial that for any $\alpha \in \mathbb{R}^m$, we have $\alpha \in \text{dom}(g)$ and $AS\alpha \in \text{dom}(f)$. Hence, all conditions to apply strong Fenchel duality results hold (Theorem 31.2, [22]). Using the fact that $g^*(y) = \frac{1}{2\lambda}y^\top(S^\top S)^\dagger y$, we get

$$\inf_\alpha f(AS\alpha) + \frac{\lambda}{2}\|S\alpha\|_2^2 = \sup_y -f^*(y) - \frac{1}{2\lambda}\|P_S A^\top y\|_2^2,$$

and the supremum is attained for some $y^* \in \text{dom}(f^*)$. The uniqueness of $y^*$ follows from strong concavity of the dual objective, which comes from the $\frac{1}{\mu}$-strong convexity of $f^*$.

We establish the existence of a minimizer $\overline{\alpha}$ of $\alpha \mapsto f(AS\alpha) + \frac{\lambda}{2}\|S\alpha\|_2^2$. The latter function is strongly convex over the subspace $(\text{Ker}S)^\perp$. Thus, there exists a unique minimizer $\overline{\alpha}$ over $(\text{Ker}S)^\perp$. Then, for any $\alpha \in \mathbb{R}^m$, writing $\alpha = \alpha_\perp + \alpha_\|$ where $\alpha_\perp \in \text{Ker}(S)^\perp$ and $\alpha_\| \in \text{Ker}(S)$, we have

$$f(AS\alpha) + \frac{\lambda}{2}\|S\alpha\|_2^2 = f(AS\alpha^\perp) + \frac{\lambda}{2}\|S\alpha^\perp\|_2^2$$
$$\geqslant f(AS\overline{\alpha}) + \frac{\lambda}{2}\|S\overline{\alpha}\|_2^2.$$

Thus, the point $\overline{\alpha}$ is a minimizer.

Let $\alpha^*$ be any minimizer. The Karush-Kuhn-Tucker conditions (Theorem 31.3, [22]) imply that $AS\alpha^* \in \partial f^*(y^*)$. Since $\partial f^* = (\nabla f)^{-1}$ (Theorem 23.5, [22]), it follows that $y^* = \nabla f(AS\alpha^*)$.

## C.3 Proof of Proposition 3 – Numerical conditioning of the re-scaled sketched program

The condition number $\kappa_\dagger$ of the re-scaled sketched program is equal to

$$\kappa_\dagger = \frac{\sup_\alpha \lambda + \sigma_1\left(\mathcal{I}_S^\top A^\top \nabla^2 f(AS\alpha)\mathcal{I}_S\right)}{\inf_\alpha \lambda + \sigma_m\left(\mathcal{I}_S^\top A^\top \nabla^2 f(AS\alpha)\mathcal{I}_S\right)},$$

where $\mathcal{I}_S = S(S^\top S)^{-\frac{1}{2}}$.

In order to show that $\kappa_\dagger \leqslant \kappa$, it suffices to upper bound the numerator in the definition of $\kappa_\dagger$ by the numerator of $\kappa$ and to lower bound the denominator of $\kappa_\dagger$ by the denominator of $\kappa$, i.e., it suffices to

show that

$$\sup_\alpha \sigma_1 \left( \mathcal{I}_S^\top A^\top \nabla^2 f(AS\alpha) A \mathcal{I}_S \right) \leqslant \sup_x \sigma_1 \left( A^\top \nabla^2 f(Ax) A \right),$$

$$\inf_\alpha \sigma_m \left( \mathcal{I}_S^\top A^\top \nabla^2 f(AS\alpha) A \mathcal{I}_S \right) \geqslant \inf_x \sigma_d \left( A^\top \nabla^2 f(Ax) A \right).$$

By the trivial inclusion $\{ S\alpha \mid \alpha \in \mathbb{R}^m \} \subseteq \mathbb{R}^d$, it holds that

$$\sup_\alpha \sigma_1 \left( \mathcal{I}_S^\top A^\top \nabla^2 f(AS\alpha) A \mathcal{I}_S \right) \leqslant \sup_x \sigma_1 \left( \mathcal{I}_S^\top A^\top \nabla^2 f(Ax) A \mathcal{I}_S \right),$$

$$\inf_\alpha \sigma_m \left( \mathcal{I}_S^\top A^\top \nabla^2 f(AS\alpha) A \mathcal{I}_S \right) \geqslant \inf_x \sigma_m \left( \mathcal{I}_S^\top A^\top \nabla^2 f(Ax) A \mathcal{I}_S \right).$$

Therefore, to establish that $\kappa_\dagger \leqslant \kappa$, it is sufficient to show that for any $x \in \mathbb{R}^d$,

$$\sigma_1 \left( \mathcal{I}_S^\top A^\top \nabla^2 f(Ax) A \mathcal{I}_S \right) \leqslant \sigma_1 \left( A^\top \nabla^2 f(Ax) A \right),$$

$$\sigma_m \left( \mathcal{I}_S^\top A^\top \nabla^2 f(Ax) A \mathcal{I}_S \right) \geqslant \sigma_d \left( A^\top \nabla^2 f(Ax) A \right).$$

The first inequality follows from the fact that $\| \mathcal{I}_S \|_2 \leqslant 1$. Hence, we have

$$
\begin{aligned}
\sigma_1 \left( \mathcal{I}_S^\top A^\top \nabla^2 f(Ax) A \mathcal{I}_S \right) &= \sup_{w \neq 0} \frac{w^\top \mathcal{I}_S^\top A^\top \nabla^2 f(Ax) A \mathcal{I}_S w}{\|w\|_2} \\
&= \sup_{w \neq 0} \frac{(\mathcal{I}_S w)^\top A^\top \nabla^2 f(Ax) A (\mathcal{I}_S w)}{\|\mathcal{I}_S w\|_2} \underbrace{\frac{\|\mathcal{I}_S w\|_2}{\|w\|_2}}_{\leqslant 1} \\
&\leqslant \sup_{z \neq 0} \frac{z^\top A^\top \nabla^2 f(Ax) A z}{\|z\|_2} \\
&= \sigma_1 \left( A^\top \nabla^2 f(Ax) A \right).
\end{aligned}
$$

For the second inequality, we distinguish two cases.

If the sketching matrix $S \in \mathbb{R}^{d \times m}$ is full-column rank, then, the matrix $\mathcal{I}_S$ is actually an isometry, i.e., for any $w \in \mathbb{R}^m$, we have $\| S(S^\top S)^{-\frac{1}{2}} w \|_2 = \|w\|_2$, which implies that

$$
\begin{aligned}
\sigma_m \left( \mathcal{I}_S^\top A^\top \nabla^2 f(Ax) A \mathcal{I}_S \right) &= \inf_{w \neq 0} \frac{w^\top \mathcal{I}_S^\top A^\top \nabla^2 f(Ax) A \mathcal{I}_S w}{\|w\|_2} \\
&= \inf_{w \neq 0} \frac{(\mathcal{I}_S w)^\top A^\top \nabla^2 f(Ax) A (\mathcal{I}_S w)}{\|\mathcal{I}_S w\|_2} \underbrace{\frac{\|\mathcal{I}_S w\|_2}{\|w\|_2}}_{=1} \\
&\geqslant \inf_{z \neq 0} \frac{z^\top A^\top \nabla^2 f(Ax) A z}{\|z\|_2} \\
&= \sigma_d \left( A^\top \nabla^2 f(Ax) A \right).
\end{aligned}
$$

Suppose now that the sketching matrix $S \in \mathbb{R}^{d \times m}$ is not full column-rank. By assumption, $S = A^\top \widetilde{S}$ where $\widetilde{S} \in \mathbb{R}^{n \times m}$ is Gaussian iid, hence, full-column rank almost surely. It implies that there exists a vector $v \neq 0$ such that $Av = 0$. Indeed, let $v \neq 0$ be a vector such that $S^\top v = 0$, which exists since $m < d$. The equation $S^\top v = 0$ can be rewritten as $\widetilde{S}^\top A v = 0$. Since $\widetilde{S}^\top$ is full row-rank, we get that $Av = 0$, i.e., $\mathrm{Ker} A \neq \{0\}$.

From $\mathrm{Ker} A \neq \{0\}$, we get $\sigma_d \left( A^\top \nabla^2 f(Ax) A \right) = 0$ and $\sigma_m \left( \mathcal{I}_S^\top A^\top \nabla^2 f(Ax) A \mathcal{I}_S \right) = 0$, which concludes the proof.

# D Proof of bounds in Table 1

## D.1 Adaptive Gaussian sketching

From Corollary 1, we have that for a target rank $k$ and a sketching dimension $m_A = 2k$, with probability at least $1 - 12e^{-k}$,

$$\frac{\|\widetilde{x} - x^*\|_2}{\|x^*\|_2} \lesssim \lambda^{-\frac{1}{2}} \left( \nu_k + \frac{1}{k} \sum_{j=k+1}^{\rho} \nu_j \right)^{\frac{1}{2}}.$$

For a matrix $A$ with rank $\rho \ll \min(n, d)$, with $k \geqslant \rho + 1$, the right hand side of the latter equation is equal to 0. In order to achieve this with probability at least $1 - \eta$, it is sufficient to oversample by an amount $\log(12/\eta)$, that is, $m_A = \rho + 1 + \log(12/\eta)$ is sufficient to achieve a $(\varepsilon = 0, \eta)$-guarantee.

For a $\kappa$-exponential decay with $\kappa > 0$, we have $\nu_j \sim e^{-\kappa j}$, and

$$\left( \nu_k + \frac{1}{k} \sum_{j=k+1}^{\rho} \nu_j \right)^{\frac{1}{2}} \sim e^{-\kappa(k+1)/2},$$

and it is sufficient for the sketching dimension $m_A$ to satisfy

$$m_A \gtrsim \kappa^{-1} \log \left( \frac{1}{\lambda \varepsilon} \right) + \log \left( \frac{12}{\eta} \right).$$

For a $\beta$-polynomial decay with $\beta > 1/2$, we have $\nu_j \sim j^{-2\beta}$ and

$$\left( \nu_k + \frac{1}{k} \sum_{j=k+1}^{\rho} \nu_j \right)^{\frac{1}{2}} \sim k^{-\beta},$$

and it is sufficient to have

$$m_A \gtrsim \lambda^{-\frac{1}{2\beta}} \varepsilon^{-\frac{1}{\beta}} + \log \left( \frac{12}{\eta} \right).$$

## D.2 Oblivious Gaussian sketching

For a $\rho$-rank matrix $A$, it has been shown in [32] that, provided the sketching dimension $m_O$ satisfies

$$m_O \gtrsim \frac{(\rho + 1) \log (2\rho/\eta)}{\varepsilon^2},$$

then,

$$\frac{\|\widetilde{x} - x^*\|_2}{\|x^*\|_2} \lesssim \varepsilon,$$

with probability at least $1 - \eta$, for any $\varepsilon \in (0, \frac{1}{2})$.

We now justify the bounds for the $\kappa$-exponential and $\beta$-polynomial decays. Let $\overline{\rho}$ be the effective rank of the matrix $AA^\top$, defined as

$$\overline{\rho} = \sum_{i=1}^{\rho} \frac{\nu_i}{\lambda + \nu_i}.$$

In [32], the authors have shown that, provided

$$m_O \gtrsim \frac{\overline{\rho}}{\varepsilon^2(\lambda + 1)} \log \left( \frac{2d}{\eta} \right),$$

then the relative error satisfies

$$\frac{\|\widetilde{x} - x^*\|_2}{\|x^*\|_2} \lesssim \varepsilon \left( 1 + \sqrt{\frac{\lambda}{\nu_k}} \right),$$

with probability at least $1 - \eta$, under the additional condition that the minimizer $x^*$ lies in the subspace spanned by the top $k$-left singular vectors of $A$. For simplicity of comparison, we neglect the latter (restrictive) requirement on $x^*$, and the term $\sqrt{\lambda/\nu_k}$ in the latter upper bound, which yields a smaller lower bound on a sufficient sketching size $m_O$ to achieve a $(\varepsilon, \eta)$-guarantee. Based on those simplifications, oblivious Gaussian sketching yields a relative error such that

$$\frac{\|\widetilde{x} - x^*\|_2}{\|x^*\|_2} \leqslant \varepsilon,$$

with probability at least $1 - \eta$, provided that $m_O \geqslant \bar{\rho}\varepsilon^{-2} \log\left(\frac{2d}{\eta}\right)$.

For a $\kappa$-exponential decay, it holds that

$$\bar{\rho} = \sum_{i=1}^{\rho} \frac{e^{-\kappa i}}{e^{-\kappa i} + \lambda} \geqslant \int_1^{\rho} \frac{e^{-\kappa t}}{e^{-\kappa t} + \lambda} dt = \int_{e^{-\kappa\rho}}^{e^{-\kappa}} \frac{1}{\kappa} \frac{1}{u + \lambda} du = \frac{1}{\kappa} \log\left(\frac{e^{-\kappa} + \lambda}{e^{-\kappa\rho} + \lambda}\right).$$

Since $\lambda \in (\nu_\rho, 1) = (e^{-\kappa\rho}, 1)$, it follows that

$$\bar{\rho} \gtrsim \kappa^{-1} \log\frac{1}{\lambda}.$$

Hence, their theoretical predictions state that the sketching dimension $m_O$ must be greater than

$$m_O \gtrsim \kappa^{-1}\varepsilon^{-2} \log\left(\frac{1}{\lambda}\right) \log\left(\frac{2d}{\eta}\right)$$

in order to achieve a $(\varepsilon, \eta)$-guarantee.

For a $\beta$-polynomial decay, it holds that

$$\bar{\rho} = \sum_{i=0}^{\rho} \frac{1}{1 + \lambda i^{2\beta}} \geq -1 + \int_0^{\rho} \frac{1}{1 + \lambda t^{2\beta}} dt = -1 + \frac{\lambda^{-1/2\beta}}{2\beta} \int_0^{\lambda\rho^{2\beta}} \frac{u^{\frac{1}{2\beta}-1}}{1 + u} du$$

$$\geq -1 + \frac{\lambda^{-1/2\beta}}{2\beta} \int_0^1 \frac{u^{\frac{1}{2\beta}-1}}{1 + u} du,$$

where the last inequality is justified by the fact that the integrand is non-negative, and the fact that $\lambda \geqslant \rho^{-2\beta}$. Since the integral is finite and independent of $\lambda$, it follows that

$$\bar{\rho} \gtrsim \lambda^{-\frac{1}{2\beta}},$$

and the sketching dimension $m_O$ must satisfy

$$m_O \gtrsim \lambda^{-\frac{1}{2\beta}}\varepsilon^{-2} \log\left(\frac{2d}{\eta}\right)$$

in order to achieve a $(\varepsilon, \eta)$-guarantee, according to their theoretical predictions.

### D.3 Leverage score column sampling.

Let $A = U\Sigma V^\top$ be a singular value decomposition of the matrix $A$, where $\Sigma = \text{diag}(\sigma_1, \sigma_2, \ldots, \sigma_\rho)$, and $\sigma_1 \geqslant \sigma_2 \geqslant \ldots \geqslant \sigma_\rho$. For a given target rank $k$, let $u_1, \ldots, u_k$ be the first $k$ columns of the matrix $U$, and denote $U_1 = [u_1, \ldots, u_k] \in \mathbb{R}^{n \times k}$. For $j = 1, \ldots, n$, define $p_j = k^{-1}\|U_{1,j}\|_2^2$, where $U_{1,j}$ is the $j$-th row of the matrix $U_1$. By orthonormality of the family $(u_1, \ldots, u_k)$, it holds that $\sum_{j=1}^n p_j = 1$, and $p_j \geqslant 0$. The family $\{p_j\}_{j=1}^n$ is called the leverage score probability distribution of the Gram matrix $AA^\top$.

Leverage based column sampling consists in, first, computing the exact or approximated leverage score distribution of the matrix $AA^\top$, and, second, sampling $m$ columns of $AA^\top$ from the latter probability distribution, with replacement. Precisely, the sketching matrix $S \in \mathbb{R}^{d \times m}$ is given as

$$S = A^\top RD,$$

where $R \in \mathbb{R}^{n \times m}$ is a column selecting matrix drawn according to the leverage scores, and $D \in \mathbb{R}^{m \times m}$ is a diagonal rescaling matrix, with $D_{jj} = (mp_i)^{-\frac{1}{2}}$, if $R_{ij} = 1$.

In order to compare the theoretical guarantees of adaptive Gaussian sketching and leverage-based column sampling, we assume that the leverage scores are computed exactly. Note that if this is not the case, then the sketching size increases as the quality of approximation of the leverage score distribution decreases. As our primary goal is to lower bound the ratio $m_S/m_A$, our qualitative comparison is not affected (at least not in the favor of adaptive Gaussian sketching) by this assumption.

The authors of [12] showed that given $\delta \in (0, 1]$, provided $m_S$ satisfies

$$m_S \gtrsim \delta^{-2} k \log\left(\frac{k}{\eta}\right),$$

then, with probability at least $1 - \eta$,

$$\|P_S^{\perp} A^{\top}\|_2 \leqslant \nu_k^{\frac{1}{2}} + \delta^2 \sum_{j=k+1}^{\rho} \nu_j^{\frac{1}{2}}.$$

Using the combination of the latter concentration bound with our deterministic regret bound (4) on the relative error, it follows that, under the latter condition on $m_S$, with probability at least $1 - \eta$,

$$\frac{\|\widetilde{x} - x^*\|_2}{\|x^*\|_2} \leqslant \lambda^{-\frac{1}{2}} \left( \nu_k^{\frac{1}{2}} + \delta^2 \sum_{j=k+1}^{\rho} \nu_j^{\frac{1}{2}} \right).$$

For a matrix $A$ with rank $\rho \ll \min(n, d)$, if the sketching size $m_S$ is greater than $(\rho + 1) \log\left(\frac{\rho+1}{\eta}\right)$, then the relative error satisfies an $(\varepsilon = 0, \eta)$-guarantee.

For a $\kappa$-exponential decay, we have

$$\nu_k^{\frac{1}{2}} + \delta^2 \sum_{j=k+1}^{\rho} \nu_j^{\frac{1}{2}} \sim \left(1 + \frac{2\delta^2}{\kappa}\right) e^{-\kappa(k+1)/2}.$$

Taking $\delta = 1/2$, it follows that the sketching size $m_S$ must be greater than $\kappa^{-1} \log\left(\frac{1}{\lambda \varepsilon}\right) \log\left(\frac{1}{\eta}\right)$ to satisfy

$$\frac{\|\widetilde{x} - x^*\|_2}{\|x^*\|_2} \lesssim \varepsilon$$

with probability at least $1 - \eta$.

For a $\beta$-polynomial decay (with $\beta > 1$), we have

$$\nu_{k+1} + \delta^2 \sum_{j=k+1}^{\rho} \nu_j \sim k^{-\beta} + \delta^2 \beta^{-1} k^{1-\beta},$$

and, provided $m_S \gtrsim \delta^2 k \log\left(\frac{k}{\eta}\right)$,

$$\frac{\|\widetilde{x} - x^*\|_2}{\|x^*\|_2} \leqslant \lambda^{-\frac{1}{2}} \left(k^{-\beta} + \delta^2 \beta^{-1} k^{1-\beta}\right).$$

To achieve a precision $\varepsilon$, it is sufficient to have

$$\varepsilon \geqslant \lambda^{-\frac{1}{2}} \left(k^{-\beta} + \delta^2 \beta^{-1} k^{1-\beta}\right)$$

Suppose first that we choose $\delta \lesssim k^{-\frac{1}{2}}$. Then, the latter sufficient condition becomes $\varepsilon \gtrsim \lambda^{-\frac{1}{2}} k^{-\beta}$. Hence, we need $k$ to be at least $\lambda^{-\frac{1}{2\beta}} \varepsilon^{-\frac{1}{\beta}}$, which implies $m_S \gtrsim \delta^{-2} k \log\left(\frac{k}{\eta}\right)$. Since $\delta \lesssim k^{-\frac{1}{2}}$, we get that $m_S$ must be greater than $k^2 \log(1/\eta)$, which further implies

$$m_S \gtrsim \lambda^{-\frac{1}{\beta}} \varepsilon^{-\frac{2}{\beta}} \log\left(\frac{1}{\eta}\right).$$

Now, suppose that we choose $\delta \gtrsim k^{-\frac{1}{2}}$. Write $\delta^2 = k^{-1+\gamma}$, where $\gamma > 0$. Since $\delta < 1$, we must have $\gamma \in (0,1)$. Further, we need $\beta\varepsilon \geqslant \lambda^{-\frac{1}{2}}\delta^2 k^{1-\beta} = \lambda^{-\frac{1}{2}}k^{\gamma-\beta}$. By assumption, $\beta > 1$, hence, $\gamma - \beta < 0$. Hence, the smallest value of $k$ that satisfies the latter inequality is given by

$$k = \left(\varepsilon^{-\frac{1}{\beta}}\lambda^{-\frac{1}{2\beta}}\right)^{\frac{1}{1-\frac{\gamma}{\beta}}}.$$

On the other hand, the smallest sketching size $m_S$ to achieve an $(\varepsilon, \eta)$ satisfies

$$m_S \gtrsim k^{2-\gamma}\log\left(\frac{1}{\eta}\right).$$

Plugging-in the value of $k$, we must have

$$m_S \gtrsim \left(\varepsilon^{-\frac{1}{\beta}}\lambda^{-\frac{1}{2\beta}}\right)^{\frac{2-\gamma}{1-\frac{\gamma}{\beta}}}\log\left(\frac{1}{\eta}\right).$$

Optimizing over $\gamma \in (0,1)$, we finally obtain that the best sufficient sketching size must satisfy

$$m_S \gtrsim \left(\varepsilon^{-\frac{1}{\beta}}\lambda^{-\frac{1}{2\beta}}\right)^{\min(2,\frac{\beta}{\beta-1})}\log\left(\frac{1}{\eta}\right).$$

## E   Extension to the non-smooth case

Here, we present some results to the case where the function $f : \mathbb{R}^n \to \mathbb{R}$ is proper, convex, but not necessarily smooth. We make the assumption that the function is $L$-Lipschitz, that is, for any $x, y \in \mathbb{R}^n$,

$$\|f(x) - f(y)\|_2 \leqslant L\|x - y\|_2.$$

In particular, this implies that the domain of the function $f^*$ is bounded, i.e., for any $z \in \text{dom}f^*$, it holds that $\|z\|_2 \leqslant L$.

Let $x^*$ be the solution of the primal program (1), which exists and is unique by strong convexity of the primal objective.

For a sketching matrix $S \in \mathbb{R}^{d \times m}$, the sketched primal program (2) admits a solution $\alpha^*$. Indeed, using arguments similar to the proof of Proposition 2, the sketched program is strongly convex over $\text{Ker}(S)^\perp$, and admits a unique solution $\alpha^*$ over that subspace. Further, for any $\alpha \in \mathbb{R}^m$, we can decompose $\alpha = \alpha_\perp + \alpha_\parallel$, with $\alpha_\perp \in \text{Ker}(S)^\perp$ and $\alpha_\parallel \in \text{Ker}(S)$. Then,

$$f(AS\alpha) + \frac{\lambda}{2}\|S\alpha\|_2^2 = f(AS\alpha_\perp) + \frac{\lambda}{2}\|S\alpha_\perp\|_2^2$$
$$\geqslant f(AS\alpha^*) + \frac{\lambda}{2}\|S\alpha^*\|_2^2.$$

As for the smooth case, using convex analysis arguments, we obtain that the dual program (14) has a solution $z^*$ which satisfies $z^* = \nabla f(Ax^*)$. Similarly, the sketched dual program (15) has a solution $y^*$ which satisfies $y^* = \nabla f(AS\alpha^*)$. Further, by first-order optimality conditions of $x^*$, we have $x^* = -\lambda^{-1}A^\top\nabla f(Ax^*)$, i.e., $x^* = -\lambda^{-1}A^\top z^*$.

As for the smooth case, we introduce the candidate approximate solution $\widetilde{x}$, defined as

$$\widetilde{x} = -\lambda^{-1}A^\top\nabla f(AS\alpha^*),$$

where $\alpha^*$ is any minimizer of (15).

**Theorem 4.** *For any $\lambda > 0$, it holds that*

$$\|\widetilde{x} - x^*\|_2 \leqslant \frac{6L}{\lambda}\sqrt{\sigma_1 Z_f}. \tag{28}$$

*where $\sigma_1$ is the top singular value of $A$.*

*Proof.* Let $\alpha^*$ be any minimizer of the sketched primal program. Following similar lines as in the proof of Theorem 1 (see Appendix B.1), it holds that

$$\|A^\top \Delta\|_2^2 \leq Z_f^2 \|\Delta\|_2^2 + z^{*\top} A P_S^\perp A^\top \Delta, \tag{29}$$

where $\Delta = y^* - z^*$. After applying Cauchy-Schwarz and using the definition of $Z_f$, inequality (29) becomes

$$\|A^\top \Delta\|_2^2 \leq Z_f^2 \|\Delta\|_2^2 + Z_f \|z^*\|_2 \|A^\top \Delta\|_2.$$

Using the fact that $\sqrt{w + w'} \leq \sqrt{w} + \sqrt{w'}$ with $w = Z_f^2 \|\Delta\|_2^2$ and $w' = Z_f \|z^*\|_2 \|A^\top \Delta\|_2$, along with the inequality $\|A^\top \Delta\|_2 \leq \sigma_1 \|\Delta\|_2$, we obtain

$$\|A^\top \Delta\|_2 \leq \sqrt{Z_f} \left( \sqrt{Z_f} \|\Delta\|_2 + \sqrt{\|z^*\|_2 \|\Delta\|_2 \sigma_1} \right).$$

Using the inequality $2ww' \leq w^2 + w'^2$ and the fact that $Z_f \leq \sigma_1$, it follows that

$$\|A^\top \Delta\|_2 \leqslant 2\sqrt{Z_f \sigma_1} \left( \|\Delta\|_2 + \|z^*\|_2 \right)$$

Dividing by $\lambda$ and using the fact that $\widetilde{x} - x^* = -\lambda^{-1} A^\top \Delta$, we get

$$\|\widetilde{x} - x^*\|_2 \leq \frac{2}{\lambda} \sqrt{Z_f \sigma_1} \left( \|y^* - z^*\|_2 + \|z^*\|_2 \right).$$

Using the fact that $\|y^*\|_2, \|z^*\|_2 \leqslant L$, we obtain the desired inequality (28). $\qquad\square$

As for the smooth-case, high-probability bounds follow from the previous deterministic bound on the relative error.