[Reviews · NeurIPS 2019]

Reviewer 1



Post-rebuttal update: The author's rebuttal addresses my (minor) concerns well, and my overall score remains the same. ---- This paper presents a principled randomized optimization method for high-dimensional convex optimization via a data-dependent random linear sketch. The approach is similar to earlier work such as: - M. Pilanci and M. J. Wainwright. Randomized sketches of convex programs with sharp guarantees. IEEE Transactions on Information Theory, 61(9):5096–5115, 2015. - Y. Yang, M. Pilanci, M. J. Wainwright, et al. Randomized sketches for kernels: Fast and optimal nonparametric regression. The Annals of Statistics, 45(3):991–1023, 2017. that also solve a sketched version convex optimization problems. The main innovations here are to extend this sketching technique to a wider class of convex objectives and to introduce a data-adaptive sketching technique that greatly improves the error bounds on the solution relative to a data-oblivious sketch. The proposed technique can also be performed iteratively to improve the accuracy of the solution without having to change the sketch matrix, so the sketch on the data only has to be performed once. Overall, I thought this was a high-quality paper. The proposed sketching approach has clear practical benefits and comes with strong theoretical guarantees. The adaptive sketch approach is particularly appealing, and has clear benefits over an oblivous sketch, and may be of broader interest to the optimization community. The paper is generally well-written and clear. I only have some minor comments, given below: I think the title should be changed to “High-dimensional Convex Optimization in Adaptive Random Subspaces” (i.e., add the word “Convex”) to better reflect the contents of the paper. Namely, the approach hinges crucially on convex duality so it seems limited in applicability to convex optimization problems. One limitation is that the optimization model (1) studied in this work assumes the loss function f is strongly smooth, with rules out L1 losses among others. Also, the model (1) can only accommodate L2 regularization of x, and not other common types of non-smooth regularization used in regression settings, e.g., L1 in the lasso or L1+L2 in the elastic net. Some discussion about whether the present approach could be extended to this more general case would be nice. I see there is a section E in the appendix that addresses this some, but this section does not appear to be discussed at all in the main body. At the start of Section 4, the paper states “By the representer theorem, the primal program (1) can be re-formulated as…”, but I do not think invoking the representer theorem here is necessary. If I understand correctly, the equation in (11) is just due to a linear change of variables x = A^T w. The results on classification of CIFAR10 in Table 3 are far from state-of-the-art (~50% classification error). Is there are different problem setting beyond classification for which the method could be demonstrated? The notation 5.10^{-5} is a little non-standard. Maybe use a \cdot or \times rather than a period?

Reviewer 2



This paper proposes a new randomized optimization method for solving high-dimensional problems. The paper is well written and the authors demonstrate the efficacy of their methods with strong theoretical guarantees and extensive experiments to back up their claims. The authors use standard results from Fenchel duality of the dual problem and its adaptively sketched counterpart to bound the relative error with high probability. The relative error bounds are shown for high dimensional matrix with finite rank, exponentially and polynomially decaying eigenvalues. The adaptive sketching method significantly outperforms its oblivious sketching counterpart for different decay values. The sketched matrix is computed exactly once and with probability 1 achieves better condition number. However, computing it involves a matrix inversion step that needs m^3 computations, is there a way to approximate it for problems which might need large m. Moreover, will the approximation still satisfy Proposition 3 upto a certain multiplicative factor? On the contrary, for small m, will dynamically modifying the sketching matrix lead to even tighter bounds? While the experiments indicate that sketch based methods indeed converge quickly to their optimal solutions, it is unclear if SGD (without sketching) converges to the same solution or if it performs better (albeit after a longer time). It will be interesting to compare the test accuracy vs wall-clock time of MNIST and CIFAR10 with oblivious sketching and other methods The authors claim that the sketch method did not work well with SGD, did the authors experiment with relatively large mini-batch sizes which may resolve the issue of not approaching the minimizer accurately. Post rebuttal: In light of the rebuttal posted by the authors, I would like to keep my current score.

Reviewer 3



Update after rebuttal: Thanks for clarifications on SGD and other baselines. Please make sure to add these experimental results in future versions of the paper. ---------------------------------------------------------------------------------------------------------- Most of my comments are on the empirical results: 1) Experiments in Section 5: While the experimental results in Section 5 suggest that the proposed technique is fast compared to direct optimization of the original objective, it is not clear how much it improves over the baselines such as oblivious Gaussian sketching. Since classification accuracy is used as the error metric (and not parameter recovery error), it could be possible oblivious sketching will have similar performance as adaptive sketching. So the authors should compare their approach with other sketching baselines and Nystrom method. The authors also point out that SGD didn't work well for adaptive sketching. This sounds slightly concerning because, for really high dimensional problems, where one has to use a large sketching dimension, Newtons method would be slower than SGD. So, it'd be great if the authors perform more experiments on the robustness of other baselines and the proposed method to the choice of optimizer used. What value of lambda is used for synthetic experiments in Figure 1? It looks like lambda = 0 is used. If this is the case, logistic regression on separable data wouldn't have a minimizer (the norm of its iterates diverges). So shouldn't the relative error keep diverging as well? 2) While the theoretical results look interesting, it is not clear why this particular form of adaptive sketching works. So it'd be great if the authors provide some intuition for why the proposed approach works. 3) Clarity: I feel the organization of the paper could be slightly improved. Some parts in the paper which are not so important (e.g., remark 1) can be pushed to the appendix and more discussion can be added to section 4. It'd be good if the authors provide some discussion on how the proposed method compares with other techniques for accelerating kernel methods such as random fourier features, nystrom methods.

[Author Response · NeurIPS 2019]

We would like to thank the reviewers for their detailed and insightful comments.

**[Performance of SGD and robustness of optimization algorithms.]** We have resolved the concerns with SGD. By
increasing the batch size towards the last iterations and averaging the last iterates, SGD on the adaptive Gaussian sketch
problem performs better in terms of time vs accuracy performance, compared to SGD on problem (1) or on the oblivious
Gaussian sketch problem. We have similar results with Adam. As reported in the submission, SVRG on the adaptive
Gaussian sketch performs better than SVRG on problem (1), and is robust to the choice of hyperparameters. Further,
Sub-sampled Newton (with mini-batch Hessian and full-batch gradient) has a strong time vs accuracy performance on
adaptive Gaussian sketch. In the revised version, we will include our new results for SGD and Adam, and a sensitivity
analysis to sketching, batch and step sizes, for all algorithms applied to the sketched problems (adaptive and oblivious).

**[Comparison with other sketching baselines.]** We carried out extensive numerical evaluations of oblivious Gaussian
sketching and adaptive sketching with uniform column sub-sampling matrix (Nystrom method) on MNIST and CIFAR10.
For a wide range of values of sketching size $m$ and regularization parameter $\lambda$, adaptive Gaussian sketching always
strongly beats oblivious sketching, and, outperforms Nystrom method, both in terms of final test accuracy (see Table 1
below), and, time vs accuracy performance for the following algorithms: SGD, SVRG, Sub-sampled Newton and Adam.
Further, adaptive Gaussian sketching matches the performance of $x^*$ for relatively small values of $m$. We will include
all these results in the revised version. **[Computational issues with $(S^\top S)^{-\frac{1}{2}}$ for large $m$.]** Thanks to this question,

Table 1: Test classification error on MNIST and CIFAR10, for 10-classes classification. "AG": Adaptive Gaussian
sketch, "Ob": Oblivious Gaussian sketch, "N": Nystrom method, $x_m$: solution obtained from problem (2) with sketching
size $m$. We mapped MNIST (resp. CIFAR10) images through 10000 (resp. 60000) random cosines.

| $\lambda$ | $x^*_{MNIST}$ | $x^{AG}_{256}$ | $x^{AG}_{1024}$ | $x^{Ob}_{256}$ | $x^{Ob}_{1024}$ | $x^{N}_{256}$ | $x^{N}_{1024}$ | $x^*_{CIFAR}$ | $x^{AG}_{256}$ | $x^{AG}_{1024}$ | $x^{Ob}_{1024}$ | $x^{N}_{256}$ | $x^{N}_{1024}$ |
|---|---|---|---|---|---|---|---|---|---|---|---|---|---|
| $5 \cdot 10^{-5}$ | 4.6 % | 4.0 % | 4.5 % | 25.2 % | 8.5% | 5.0 % | 4.6 % | 51.6 % | 50.6% | 51.0% | 70.5% | 55.8% | 53.1% |
| $5 \cdot 10^{-6}$ | 2.5% | 2.8% | 2.4% | 30.1% | 9.4% | 3.0% | 2.7% | 47.6% | 51.9% | 45.8% | 80.1% | 57.2% | 55.8% |

16
we have improved our results and we can show that the matrix $(S^\top S)^{-\frac{1}{2}}$ can be replaced by any other pre-conditioner
$Q$, and in particular, for large $m$, a matrix $Q$ obtained by approximate SVD. Provided $\|Q - (S^\top S)^{-\frac{1}{2}}\|_2$ is small, then
the condition number of (7) remains close to that of (1). Importantly, it does not affect any of the bounds on $\widetilde{x}$. We will
include these results in the revised version.

**[For small $m$, would dynamically modifying the sketching matrix lead to tighter bounds?]** For small $m$, we tried
numerically to refresh the sketching matrix at each iteration and it did not yield good results. However, our Algorithm 2
refreshes the sketching matrix at the end of each optimization, and gives tighter bounds.

**[Results on CIFAR10 far from state-of-the-art. Other optimization problems for which the method could be**
**demonstrated?]** We did additional experiments with features extracted from a pre-trained neural network, and $\widetilde{x}$
matched exactly the test error of $x^*$ ($\sim 10\%$). We will include these results in the final version. Beyond classification,
large-scale generalized linear models (other than least squares) can be addressed with our method.

**[Unclear if SGD (without sketching) converges to the same solution or performs better].** The reported results
correspond to the best SGD solution we obtained (with grid search of the batch and step sizes), even through longer
time horizons.

**[Value of $\lambda$ used for synthetic experiments?]** We used $\lambda = 10^{-4}$. Thank you for pointing this out, we will fix it.

**[Title suggestion.]** We agree with the relevant title suggestion. **[Non-convexity].** We derived a new result regarding
non-convex, smooth functions $f$: if $\alpha^*$ is a nearly-stationary point for the sketched problem (2), then $\widetilde{x}$ is a nearly
$\varepsilon$-stationary point for problem (1), where $\varepsilon$ controlled again by the tail spectral decay of $A$. **[Regularity assumptions**
**on $f$].** We will add some discussion in the main body of the paper. In Appendix E, guarantees are provided for convex,
non-smooth objectives $f$. **[Other regularizers.]** We have extended our analysis to smooth, strongly convex regularizers.
However, extension to the $L_1$-norm is an open question.

**[Invoking the represter theorem not necessary in Eq. (11).][Notation $5.10^{-5}$ non-standard.]** We will simplify
the argument for Eq. (11) in the revised version, and correct the notation. Thank you for pointing this out.

**[Intuition for why the proposed approach works.]** We will discuss more carefully some intuition in the revised
version. In a nutshell, the kernelized version of optimization problem (1) is well approximated by $\min_w f(AP_S A^\top w) +$
$\lambda \|P_S A^\top w\|^2$, provided that $AA^\top \approx AP_S A^\top$. Adaptive sketching works better than oblivious one, since $\|AA^\top -$
$AP_{A^\top \widetilde{S}} A^\top\|_2 \ll \|AA^\top - AP_{\widetilde{S}} A^\top\|_2$, for $\widetilde{S}$ i.i.d. Gaussian.

**[Comparison of the proposed method with approximate kernel methods]** Random Fourier features lead to problems
of type (1). Standard Nystrom methods approximates the matrix $K$ in (11). But both problems (1) and (11) are typically
high-dimensional. Our method is a dimension-reduction tool, that can be used on top of approximate kernel methods.

[Meta-Review · NeurIPS 2019]

The paper has solid theoretical contributions and is written well. The following are some points that came up during discussions: 1. An additional reviewer pointed out that the proposed adaptive sketch appears to be analogue of leverage score sampling with iid Gaussian sketch - the idea is similar to the intuitions the authors provide in the rebuttal about finding a good approximation of AA^T with AP_SA^T. Although this connection at this point is not rigorously worked out, it would be very useful if the authors can add discussions/connections about this in their revision. The connection could also explain the similarity of the leverage score rates in Table 1. 2. Although the paper was primarily evaluated on theoretical grounds, initial reviews had raised some concerns on the empirical evaluation. The author response provides more elaborate empirical results. Please include these results and discussion in the final version. 3. The following missing reference on sketching for solving system of equations is relevant: https://arxiv.org/pdf/1506.03296.pdf